# xDial-Eval: A Multilingual Open-Domain Dialogue Evaluation Benchmark

**Chen Zhang**[†]    **Luis Fernando D'Haro**[‡]    **Chengguang Tang**[★]

**Ke Shi**[★]    **Guohua Tang**[★]    **Haizhou Li**[♥,†]

[†]National University of Singapore, Singapore

[‡]Universidad Politécnica de Madrid, Spain    [★]Tencent AI Lab, China

[♥]Shenzhen Research Institute of Big Data, School of Data Science,
The Chinese University of Hong Kong, Shenzhen, China

chen_zhang@u.nus.edu

## Abstract

Recent advancements in reference-free learned metrics for open-domain dialogue evaluation have been driven by the progress in pre-trained language models and the availability of dialogue data with high-quality human annotations. However, current studies predominantly concentrate on English dialogues, and the generalization of these metrics to other languages has not been fully examined. This is largely due to the absence of a multilingual dialogue evaluation benchmark. To address the issue, we introduce xDial-Eval, built on top of open-source English dialogue evaluation datasets. xDial-Eval includes 12 turn-level and 6 dialogue-level English datasets, comprising 14930 annotated turns and 8691 annotated dialogues respectively. The English dialogue data are extended to nine other languages with commercial machine translation systems. On xDial-Eval, we conduct comprehensive analyses of previous BERT-based metrics and the recently-emerged large language models. Lastly, we establish strong self-supervised[1] and multilingual baselines. In terms of average Pearson correlations over all datasets and languages, the best baseline outperforms OpenAI's ChatGPT by absolute improvements of 6.5% and 4.6% at the turn and dialogue levels respectively, albeit with much fewer parameters. The data and code are publicly available at https://github.com/e0397123/xDial-Eval.

## 1 Introduction

Open-domain dialogue evaluation is a long-lasting challenge to dialogue system research (Mehri et al., 2022a). Currently, human evaluation is the most reliable way to holistically judge the quality of the dialogue. Due to the high costs and low reproducibility of human evaluation, automatic metrics are proposed to complement it.

There are two distinct paradigms of automatic evaluation, reference-based, and reference-free. Reference-based metrics, such as BLEU (Papineni et al., 2002) and Embedding Average (Mitchell and Lapata, 2008) are widely adopted in the research community due to their easy implementation and general applicability to various types of dialogues. Yet, they can be misleading due to the poor correlation with human judgment (Liu et al., 2016; Mehri et al., 2022a). On the other hand, reference-free metrics bypass the reliance on references and directly estimate the quality of a single response (turn level) or a multi-turn dialogue (dialogue level).

Currently, there's a growing interest in creating model-based, reference-free metrics. One line of work focuses on learning a discriminative metric with self-supervised learning - a model is trained to distinguish high-quality responses/dialogues from low-quality responses/dialogues based on weak supervision signals that are automatically constructed from human-human dialogue data (Yeh et al., 2021). These metrics benefit from the BERT-based language models (Devlin et al., 2019; Liu et al., 2019) and the availability of high-quality dialogue corpora (Li et al., 2017; Zhang et al., 2018). With the recent advancement of large language models (LLMs) (Brown et al., 2020; Touvron et al., 2023a; Ouyang and et al., 2022; OpenAI, 2023), an emerging line of work is to treat the LLMs as annotators, which judge the quality of responses/dialogues through prompting (Gupta et al., 2022; Huynh et al., 2023; Fu et al., 2023). Different from the BERT-based metrics, such metrics are generative in nature.

A common attribute of both metric categories is that they are not trained on dialogue evaluation data with human quality annotations, yet they exhibit significant potential in simulating how humans perform evaluations at the turn or dialogue level. Despite the significant progress in the field, current research predominantly focuses on the En-

---

[1]All methods examined in the paper are not trained on data with human quality annotations. xDial-Eval is purely used for testing purposes.

glish language, other languages receive insufficient attention. To expand the scope of automatic dialogue evaluation research beyond English and thoroughly investigate the language generalization potential of model-based reference-free metrics, we propose a large-scale multilingual open-domain dialogue evaluation benchmark, named xDial-Eval. To construct xDial-Eval, we curate 12 turn-level and 6 dialogue-level open-source English evaluation datasets. In total, the turn-level and dialogue-level datasets comprise 14930 human-annotated turns and 8691 human-annotated multi-turn dialogues respectively. Then, we translate the English data into nine different languages with commercial machine translation models.

In addition, we comprehensively assess the performance of the two metric categories on xDial-Eval. Pertaining to the discriminative category, previous state-of-the-art (SoTA) BERT-based methods are analyzed, while for the generative category, we evaluate the multilingual dialogue evaluation capability of recent LLMs, especially the instruction-tuned variants, such as ChatGPT[2], Alpaca (Taori et al., 2023), and Vicuna (Chiang et al., 2023). In our analysis, we systematically explore the effects of training data, training strategies, and pretrained models on multilingual dialogue evaluation.

Lastly, we introduce strong multilingual self-supervised baselines on xDial-Eval. Specifically, we fine-tune the LLMs on synthetic instruction data built from human-human dialogues. The fine-tuned models are found to exhibit much stronger multilingual dialogue evaluation capability than the original LLMs. Motivated by the complementary nature of generative and discriminative metrics, we perform metric ensemble, which yields strong correlations with human evaluation and language generalization capability on xDial-Eval, even outperforming the powerful ChatGPT, which has been recently proven to be a superior reference-free text evaluator (Chen et al., 2023a; Liu et al., 2023).

## 2 Related Work

A long-lasting goal of automatic dialogue evaluation is to fully approximate human evaluation, which is quantified by strong correlations (0.8+) with ground-truth human quality annotations (Mehri et al., 2022a). Recent research is dedicated to developing powerful model-based reference-free metrics for automatic dialogue evaluation (Yeh et al., 2021).

**BERT-Based Discriminative Metrics** - A series of works (Sai et al., 2020; Huang et al., 2020; Sinha et al., 2020; Lan et al., 2020; Mehri and Eskenazi, 2020b; Phy et al., 2020; Pang et al., 2020; Zhang et al., 2021c) targets turn-level evaluation and leverages self-supervised learning. They rely on negative sampling strategies, such as random utterance replacement and word order shuffling, to generate synthetic data for training discriminative models. Another group of metrics is learned to holistically judge the quality of multi-turn dialogues (Mesgar et al., 2020; Zhang et al., 2021a; Ghazarian et al., 2022b; Zhang et al., 2022a) with a similar idea that a model is trained to distinguish original human-human dialogues from negative samples generated by strategies, such as utterance order shuffling.

All the metrics rely on either BERT (Devlin et al., 2019) or RoBERTa (Liu et al., 2019) for dialogue context understanding and modeling the utterance-level interactions. Although they exhibit good correlations with human evaluation on English benchmark datasets, their transferability across different languages has not been explored. In this study, we select representative discriminative metrics and conduct correlation analyses on the multilingual xDial-Eval benchmark.

**LLM-Based Generative Metrics** - The evolution of large language models (LLM) has drastically changed the NLP landscape (Brown et al., 2020; Chowdhery et al., 2022; Muennighoff et al., 2023; Touvron et al., 2023a). Especially, the line of works on instruction-tuning of LLMs (Ouyang and et al., 2022; Bai et al., 2022; Chung et al., 2022; OpenAI, 2023; Taori et al., 2023; Chiang et al., 2023; Chen et al., 2023b) has led to general-purpose AI assistants that exhibit remarkable ability in understanding and following users' instructions.

In the context of automatic dialogue evaluation, these LLMs can serve as unified evaluators of both the response and the multi-turn dialogue quality through prompting with task-specific templates. For example, Gupta et al. (2022) introduces InstructDial, which consists of 48 diverse dialogue tasks in a unified text-to-text format. Models tuned on the InstructDial dataset demonstrate good zero-shot performance in the dialogue evaluation task. Huynh et al. (2023) conduct a comprehensive analysis of the dialogue evaluation capability of LLMs with varying model types, sizes, choices of

---

[2] https://openai.com/blog/chatgpt

training data, etc. They observe that the smaller language models fine-tuned on instructions and dialogue-specific data can outperform very large language models. We move beyond both works by comprehensively exploring the LLMs' ability in evaluating multilingual dialogue.

More recently, several works (Chen et al., 2023a; Liu et al., 2023; Lin and Chen, 2023) study the dialogue evaluation capability of closed-source instruction-following LLMs via prompting, such as OpenAI's ChatGPT and Anthropic Claude (Bai et al., 2022). These closed-source LLMs exhibit strong zero-shot correlations with human evaluation. Yet, a major limitation is that it is not always feasible to adapt such models to custom data. Our work serves as the first study exploring the dialogue evaluation capability of both the closed-source and the recent open-source instruction-tuned LLMs on multilingual dialogue evaluation data. Additionally, we move beyond prompt engineering by performing task-specific finetuning of the open-source LLMs to boost the dialogue evaluation ability and language generalization of such models.

**Multilingual Open-Domain Dialogue** - Existing studies on multilingual open-domain dialogue mainly target dialogue response selection and dialogue generation (Lin et al., 2021) with little emphasis on automatic dialogue evaluation. For example, Sato et al. (2018) constructs a multilingual dataset, which contains the Ubuntu IRC logs in 12 different languages for response selection. Zhang et al. (2021d) proposes MRS, a multilingual reply suggestion dataset with ten languages. More related to our work, Rodríguez-Cantelar et al. (2023) recently held the shared task on "Robust and Multilingual Automatic Evaluation Metrics for Open-Domain Dialogue Systems" at DSTC11[3]. They release a series of multilingual evaluation datasets and a multilingual deep AM-FM (Zhang et al., 2021b) baseline. Yet, their released data is only limited to 3 languages. Most of the data target turn-level evaluation with a limited amount of dialogue-level annotated data. On the contrary, our proposed benchmark, xDial-Eval, contains large-scale annotated data at both turn and dialogue levels, spanning 10 different languages.

## 3 The xDial-Eval Benchmark

### 3.1 Benchmark Details

The xDial-Eval benchmark is built on top of open-source English evaluation datasets. The statistics of the datasets are outlined in Table 1. The English datasets comprise 14930 annotated turns and 8691 annotated multi-turn dialogues. We utilize the Microsoft Azure translation service[4] to translate all the datasets into nine diverse languages: Chinese (ZH), Spanish (ES), German (DE), French (FR), Japanese (JA), Korean (KO), Hindi (HI), Arabic (AR), and Russian (RU). In total, our translation effort comprised 183,495 unique utterances and cost approximately 400 USD. We keep the original dialogue quality annotations from the English datasets for the newly translated dialogue data. Examples of the translated data are included in Appendix B.

| Turn-Level Datasets | #Instance | #Utts/Instance | #Ctx/Hyp Words | #Dims |
|---|---|---|---|---|
| Persona-USR (2020b) | 300 | 9.3 | 98.4 / 12.0 | 6 |
| Persona-Zhao (2020) | 900 | 5.1 | 48.8 / 11.5 | 4 |
| ConvAI2-GRADE (2020) | 600 | 3.0 | 24.4 / 11.3 | 1 |
| Persona-DSTC10 (2022b) | 4,829 | 4.0 | 36.0 / 11.6 | 4 |
| DailyDialog-GRADE (2020) | 300 | 3.0 | 26.0 / 10.8 | 1 |
| DailyDialog-Zhao (2020) | 900 | 4.7 | 47.5 / 11.0 | 4 |
| DailyDialog-Gupta (2019) | 500 | 4.9 | 49.9 / 10.9 | 1 |
| Topical-USR (2020b) | 360 | 11.2 | 236.3 / 22.4 | 6 |
| Topical-DSTC10 (2022b) | 4,500 | 4.0 | 50.6 / 15.9 | 4 |
| Empathetic-GRADE (2020) | 300 | 3.0 | 29.0 / 15.6 | 1 |
| FED-Turn (2020a) | 375 | 10.4 | 87.3 / 13.3 | 9 |
| ConTurE-Turn (2022a) | 1066 | 3.8 | 21.67 / 10.99 | 1 |
| **Dialogue-Level Datasets** | **#Instance** | **#Utts/Instance** | **#Words/Utt** | **#Dims** |
| IEval (2022) | 1,920 | 6.0 | 12.4 | 8 |
| Persona-See (2019) | 3,316 | 12.0 | 7.6 | 9 |
| Reliable-Eval (2022) | 2,925 | 21.2 | 8.4 | 7 |
| ConTurE-Dial (2022b) | 119 | 17.9 | 8.6 | 11 |
| FED-Dial (2020a) | 125 | 12.7 | 9.2 | 11 |
| Human-Eval (2022) | 286 | 12.0 | 11.6 | 3 |

Table 1: Statistics of English evaluation datasets in xDial-Eval. Ctx, Hyp, and Dim refer to context, hypothesis, and dimension respectively. Dimension means the annotated response /dialogue quality aspect.

### 3.2 Translation Quality Assurance

We verify the translated data quality with both automatic and human evaluation. Due to the high costs of running a full evaluation, we randomly sample 100 utterances from each translated dataset, summing up to 1700 translated utterances per non-English language and a total of 15300 translation pairs[5]. Both the automatic and human evaluation results suggest a high quality of the translated xDial-Eval data.

**Automatic Measures** Due to the absence of target language references, we apply direct qual-

---

[3] https://dstc11.dstc.community/tracks

[4] https://learn.microsoft.com/en-gb/azure/cognitive-services/translator/

[5] ConTurE-Turn & ConTurE-Dial share the same data.

ity estimation with three different models: OpenAI's GPT-4 model (OpenAI, 2023), Unbabel's wmt22-cometkiwi-da (Rei et al., 2022) and wmt23-cometkiwi-da-xl (Rei et al., 2023) models. In addition to quality estimation, we perform additional back-translation of the translated content to English using Azure MT and then conduct a reference-based evaluation comparing the English source and back-translated utterances.

For quality estimation with GPT-4, we prompt the model to evaluate the adequacy of each translation pair on a scale of 1 to 5, with 1 denotes poor adequacy and 5 denotes perfect adequacy. The instruction template is outlined in Appendix A. For the reference-based evaluation, we adopt sacreBLEU[6] (Papineni et al., 2002), BERTScore[7] (Zhang et al., 2020), and BLEURT[8] (Sellam et al., 2020) to assess the 15300 English source and back-translation pairs. The automatic evaluation results are summarized in Table 2. We can observe that the scores of the automatic metrics are generally high.

| Lang | BLEU | BERTScore | BLEURT | GPT-4 | CometKiwi22 | CometKiwi23 |
|---|---|---|---|---|---|---|
| EN-ZH | 37.85 | 0.691 | 0.773 | 4.58 | 0.827 | 0.739 |
| EN-ES | 56.58 | 0.767 | 0.820 | 4.64 | 0.845 | 0.764 |
| EN-DE | 50.30 | 0.754 | 0.817 | 4.63 | 0.834 | 0.744 |
| EN-FR | 52.10 | 0.748 | 0.811 | 4.59 | 0.842 | 0.709 |
| EN-JA | 42.70 | 0.711 | 0.786 | 4.33 | 0.854 | 0.778 |
| EN-KO | 40.23 | 0.707 | 0.782 | 4.26 | 0.846 | 0.767 |
| EN-HI | 49.33 | 0.726 | 0.797 | 4.40 | 0.793 | 0.699 |
| EN-AR | 48.54 | 0.737 | 0.794 | 4.39 | 0.835 | 0.736 |
| EN-RU | 49.13 | 0.741 | 0.804 | 4.40 | 0.825 | 0.739 |
| Avg | 47.42 | 0.731 | 0.798 | 4.47 | 0.833 | 0.742 |

Table 2: Automatic Evaluation Results. Score ranges of BLEU, BERTScore, BLEURT, GPT-4, wmt22-cometkiwi-da (CometKiwi22), and wmt23-cometkiwi-da-xl (CometKiwi23) are [0, 100], [0, 1], [0, 1], [1, 5], [0, 1], and [0, 1] respectively.

**Human Evaluation** We also conduct the human evaluation to cross-validate the translation quality. Specifically, we collaborate with a service provider to recruit native speakers, who are proficient in both English and their native language, such as Chinese, Spanish, etc. The bilingual speakers are instructed to evaluate the quality of translations from English to their mother tongue. Each speaker assesses 350 translation pairs, rating them on a 1-5 scale, where 1 indicates poor translation and 5 indicates excellent translation. The human evaluation covered nine

---

[6] https://huggingface.co/spaces/evaluate-metric/sacrebleu

[7] https://github.com/Tiiiger/bert_score.

[8] https://github.com/google-research/bleurt/blob/master/checkpoints.md

language pairs, totaling 3150 instances. Quality check on a random subset of the human evaluation data is conducted to ensure their reliability. Additionally, for the 350 English-to-Chinese translations, three authors of the paper manually evaluated the translation. The inter-annotator agreement is 0.578, indicating medium agreement. The average human evaluation scores for EN-ZH, EN-ES, EN-DE, EN-FR, EN-JA, EN-KO, EN-HI, EN-AR, and EN-RU language pairs are 4.75, 4.68, 4.59, 4.53, 4.39, 4.32, 4.17, 4.41, and 4.71 respectively. The human evaluation results agree with the automatic results, showcasing that the translation quality of xDial-Eval is good.

## 4 Model-Based Reference-Free Metrics

**Mathematical Formulation** - Let $D^{i,l}$ denote a dialogue evaluation dataset in xDial-Eval with index $i$ and language $l$. $D^{i,l}$ either comprises $N$ number of multi-turn dialogues (dialogue-level) or $J$ number of context-response pairs (turn-level). Each data instance can be either a multi-turn dialogue $d_j^{i,l}$ or a context-response pair $(c_j^{i,l}, r_j^{i,l}) \in D^{i,l}$ where $j \in \{1, ..., N\}$. Each $d_j^{i,l}$ or $r_j^{i,l}$ is evaluated by human judges using a predefined Likert scale to measure specific aspects of dialogue or response quality. Given the multifaceted nature of quality and our particular interest in the language generalization of the automatic metrics, we are concentrating our analysis on "coherence" at the dialogue level and "context relevance" at the turn level, which are the most studied dimensions in the dialogue evaluation literature (Yeh et al., 2021). We will explore other dimensions in future works.

We denote the mean quality score assigned by human annotators to a specific data instance as $h_j^{i,l}$. The metric model, $M$, assigns metric score $s_j^{i,l}$ to $d_j^{i,l}$ or $r_j^{i,l}$. The performance of $M$ on $D^{i,l}$ is assessed by computing the Pearson or Spearman correlations, $\rho^{i,l}$, between $S^{i,l} = \{s_1^{i,l}, \ldots, s_N^{i,l}\}$ and $H^{i,l} = \{h_1^{i,l}, \ldots, h_N^{i,l}\}$. The higher the $\rho^{i,l}$, the better the performance of $M$ on $D^{i,l}$. $M$ with strong evaluation capability achieves high $\frac{1}{|\Omega|} \sum_{i \in \Omega} \rho^{i,l}$ where $\Omega$ is either the collection of turn-level datasets or the collection of dialogue-level datasets in xDial-Eval. The language generalization of $M$ can be assessed by $\frac{1}{|L|} \sum_{l \in L} \left( \frac{1}{|\Omega|} \sum_{i \in \Omega} \rho^{i,l} \right)$ where $L$ is the set of languages covered by xDial-Eval.

**BERT-Based Discrminative Metrics** - In this

| ### Instruction:
Given the input dialogue between a human ([H]) and a chatbot ([C]) below, predict whether the dialogue is coherent?

### Input:
[H] My dog puked all over the carpet a few weeks back.
[C] Oh no! Did you clean it up?
[H] yes
[C] Did you clean it up?
[H] yes I did
[C] That's great! I'm glad it worked out for you!

### Response:
**No** | ### Instruction:
Given the following dialogue context and response, predict whether the response is relevant to the context.

### Input:
Dialogue Context:
hello , tell me something about yourself !
i am 56 years old and i am a housekeeper .
do you have any hobbies or interests ?

Dialogue Response:
i like to read and write poetry .

### Response:
**Yes** |
|---|---|
| (a) Dialogue-Level Evaluation | (b) Turn-Level Evaluation |

Table 3: Alpaca-7B Instruction Template. The bold text is the expected output and the rest are the input to the LLM.

study, we select two SoTA BERT-based metrics in the open-domain dialogue evaluation literature: PoE (Zhang et al., 2023) and FineD-Eval (Zhang et al., 2022a). Both are self-supervised metrics designed for turn-level and dialogue-level evaluation respectively. Their detailed descriptions are outlined in Appendix C.1. The original PoE and FineD-Eval are based on RoBERTa-Large (354M) and RoBERTa-Base (125M) (Liu et al., 2019) respectively. To adapt them for the multilingual dialogue evaluation task, we reimplement them with XLM-R-Large (550M) and XLM-R-base (270M) (Conneau et al., 2020) respectively.

To finetune the models, we need multilingual training data. The original synthetic data for training PoE comprises approximately 2M English context-response pairs while that of FineD-Eval contains roughly 90K multi-turn dialogues. Both training datasets contain a balanced number of positive and negative data instances[9]. We employ the open-source NLLB-200-3.3B machine translation model (Costa-jussà et al., 2022) to convert the English data into the other 9 languages. For easy reference, we denote the multilingual training datasets as xPoE-Turn and xFined-Dial[10]. We didn't use the high-quality Azure MT service for training data translation because the costs are exorbitant and we can also check whether training on multilingual data with lower translation quality causes a signifi-

cant negative impact on model performance.

When scoring $d_j^{i,l}$ with FineD-Eval or $r_j^{i,l}$ with PoE, the flattened token sequence of $d_j^{i,l}$ or $(c_j^{i,l}, r_j^{i,l})$ is input into FineD-Eval or PoE respectively. The scalar normalized score $s_j^{i,l}$ in the range [0, 1] is output from the models accordingly.

**LLM-Based Generative Metrics** - We select a diverse set of popular LLMs with different backbones and pretraining data. Due to the fast development pace of LLMs, the following list is not exhaustive: LLaMA-7B (Touvron et al., 2023a), LLaMA-2-7B (Touvron et al., 2023b), Baichuan-2-7B (Yang et al., 2023), Alpaca-7B (Taori et al., 2023), Vicuna-7B-V1.1 (Chiang et al., 2023), BLOOM-7B (Scao et al., 2023), Phoenix-7B (Chen et al., 2023b), Falcon-7B (Almazrouei et al., 2023), and OpenAI's ChatGPT (gpt-3.5-turbo-0301). Due to computation limitations, the larger LLMs variants are not explored in this study. The detailed descriptions of each model are included in the Appendix C.2.

To score $d_j^{i,l}$ or $r_j^{i,l}$ with the LLMs, we first need to convert the input instance to the corresponding model- and task-specific instruction-based prompt templates. Take Alpaca as an example, table 3 displays the specific prompts for both dialogue-level and turn-level evaluation tasks[11]. Following Gupta et al. (2022), we frame the evaluation task as a binary classification problem. Given an input prompt, we specifically focus on the probabilities related to the label words "Yes" and "No" as generated by the language model. Then, we

---

[9]For PoE, positive and negative refer to relevant and irrelevant responses while for FineD-Eval, they refer to coherent and incoherent dialogues.

[10]Given that xPoE-Turn and xFined-Dial sizes are 10 times their original English datasets, we sample a subset equal to the original English size for model training.

[11]Appendix D includes more example instruction templates for other models and languages.

normalize the probability of "Yes" as $P(\text{Yes}) = P(\text{Yes})/(P(\text{Yes}) + P(\text{No}))$ and $P(\text{Yes})$ serves as $s_j^{i,l}$ of $d_j^{i,l}$ or $r_j^{i,l}$. As we do not have access to the output probabilities of the closed-source Chat-GPT, we prompt ChatGPT to explicitly provide a numerical score to $d_j^{i,l}$ or $r_j^{i,l}$ on the scale of 1 to 5.

**Proposed Metrics** - A drawback of LLMs is their lack of specialized focus. While these models are constructed to function as versatile AI assistants capable of managing a variety of NLP tasks, their performance may not measure up to domain-specific experts when dealing with specialized tasks or particular domains. Hence, we propose to further adapt the open-source LLMs to custom data designed for automatic dialogue evaluation.

An additional note is that current LLM finetuning often uses human-annotated data. However, due to challenges in scaling up high-quality, human-annotated training data collection for open-domain dialogue evaluation, and the proven success of SoTA BERT-based metrics using automatically-constructed synthetic dialogue data, we propose to investigate whether LLMs can also benefit from finetuning with synthetic data.

Specifically, we reuse the xPoE-Turn and xFined-Dial datasets described in the previous section to perform instruction-tuning of the LLMs. To speed up the experiments, we sample a subset of 100k (10k per language) from each dataset. Subsequently, we transform the data into an instruction format using model-specific prompt templates, similar to those in Table 3. Then, the LLMs are finetuned on the 200K multilingual instruction data using the Low-Rank Adaptation (LoRA) technique (Hu et al., 2022). The dialogue/response scoring process of the finetuned LLMs is the same binary classification formulation discussed above.

## 5 Experiment Setup

We explore the effects on multilingual dialogue evaluation with both zero-shot prompting and fine-tuning the LLMs on different types of data[12]. For LLM finetuning, we first investigate the cross-lingual generalization of the LLMs when only finetuned on the English synthetic data. Models with different pretrained backbones are chosen for investigation: PoE, Fined-Eval, Alpaca-7B, Phoenix-7B, LLaMA-2-7B, and Baichuan-2-7B.

Secondly, we finetune the LLMs using the syn-

[12]The computation details are outlined in Appendix F.

thetic multilingual dialogue data to determine if this strategy improves their language generalization. In particular, we examine two groups of LLMs. The first is vanilla LLMs without instruction tuning, including LLaMA-7B, BLOOM-7B, LLaMA-2-7B, and Baichuan-2-7B. The second group includes the instruction-tuned variants: Alpaca-7B and Phoenix-7B. By comparing these two groups after finetuning on the multilingual synthetic data, we study whether a two-stage instruction finetuning is useful, i.e., an LLM is first finetuned on general-purpose instruction data and then further adapted to custom data. Additionally, we also want to find out which 7B open-source LLM possesses the strongest multilingual generalization.

Furthermore, we explore the ensemble of the finetuned LLMs and the BERT-based discriminative metrics and examine whether their difference in training objectives helps complement each other in their dialogue evaluation abilities. Since the finetuned LLMs and the BERT-based metrics produce scores ranging from 0 to 1 (described in §4), we achieve the ensemble by simply calculating the arithmetic mean of their respective output scores.

## 6 Results & Analysis

Table 4 and 5 present the key experiment results[13].

### 6.1 Zero-Shot Performance of LLMs

**Vanilla vs Instruction-Tuned LLMs** - Table 5's "LLMs-Zeroshot" category shows that vanilla LLMs exhibit low correlations with human evaluations across all languages. In contrast, instruction-tuned LLMs outperform their vanilla counterparts, with better average Pearson correlations at both turn and dialogue levels. For example, Alpaca-7B achieves average Pearson improvements of 0.218 and 0.336 over LLaMA-7B at turn and dialogue levels respectively. Similar improvements are seen when comparing Vicuna-7B to LLaMA-7B and Phoenix-7B to BLOOM-7B. These significant boosts in performance are credited to the closer alignment of instruction-tuned LLMs with humans' task-solving abilities.

**Impact of Backbone Models** - Baichuan-2-7B and Falcon-7B perform much better than other vanilla

[13]Full experiment results on each dataset can be found at `https://docs.google.com/spreadsheets/d/1w2PoIk2BqlkYEYiZ_LbOEGCS1qTtEVmtlz5Ex9YOQ_M/edit?usp=sharing` and Additional supporting analyses are included in Appedix E

| | | | | | Turn-Level | | | | | | |
|---|---|---|---|---|---|---|---|---|---|---|---|
| **Models** | EN | ZH | ES | DE | FR | JA | KO | HI | AR | RU | AVG |
| PoE | 0.487 | 0.444 | 0.455 | 0.459 | 0.459 | 0.436 | 0.425 | 0.359 | 0.426 | 0.440 | 0.439 |
| Alpaca-7B | 0.499 | 0.347 | 0.410 | 0.422 | 0.421 | 0.262 | 0.221 | 0.163 | 0.199 | 0.352 | 0.330 |
| Phoenix-7B | 0.439 | 0.392 | 0.377 | 0.327 | 0.417 | 0.282 | 0.201 | 0.372 | 0.398 | 0.270 | 0.348 |
| LLaMA-2-7B | 0.509 | 0.390 | 0.458 | 0.414 | 0.445 | 0.313 | 0.311 | 0.242 | 0.246 | 0.381 | 0.371 |
| Baichuan-2-7B | 0.556 | 0.513 | 0.469 | 0.468 | 0.472 | 0.385 | 0.349 | 0.256 | 0.294 | 0.416 | 0.418 |
| | | | | | Dialogue-Level | | | | | | |
| FineD | 0.376 | 0.349 | 0.354 | 0.351 | 0.356 | 0.363 | 0.347 | 0.320 | 0.310 | 0.344 | 0.347 |
| Alpaca-7B | 0.408 | 0.309 | 0.355 | 0.345 | 0.364 | 0.219 | 0.194 | 0.199 | 0.200 | 0.314 | 0.291 |
| Phoenix-7B | 0.334 | 0.342 | 0.248 | 0.238 | 0.297 | 0.246 | 0.179 | 0.277 | 0.310 | 0.202 | 0.267 |
| LLaMA-2-7B | 0.372 | 0.298 | 0.348 | 0.354 | 0.342 | 0.245 | 0.241 | 0.201 | 0.207 | 0.313 | 0.292 |
| Baichuan-2-7B | 0.359 | 0.328 | 0.289 | 0.319 | 0.322 | 0.268 | 0.269 | 0.228 | 0.256 | 0.289 | 0.293 |

Table 4: Language-wise average turn-level (over 12 datasets) and dialogue-level (over 6 datasets) Pearson correlations of models finetuned on English data only. The corresponding Spearman results can be found in Table 16.

LLMs in terms of correlation at both turn and dialogue levels, possibly due to its pretraining data's similarity with xDial-Eval benchmark data. Unlike LLaMA, pretrained on general web text like CommonCrawl and Wikipedia, or BLOOM, pretrained on 498 HuggingFace NLP datasets, Falcon-7B uses a blend of filtered web data and curated high-quality corpora like social media conversations (Penedo et al., 2023). Baichuan-2-7B is pretrained on large-scale and diverse multilingual data, totaling 2.6 trillion tokens. Alpaca-7B and Vicuna-7B generally perform better in Latin and Cyrillic languages. However, Phoenix-7B exhibits more consistent performance across languages. For instance, Alpaca-7B achieves > 0.25 average Pearson correlations in English, Spanish, German, French, and Russian, but < 0.2 in other languages at the turn level. Phoenix-7B shows less performance variation. Similar trends are observed at the dialogue level. Differences in language generalization are largely due to their backbone models and instruction-tuning data. BLOOM and Baichuan are multilingual LLMs while LLaMA and Falcon are mainly pretrained on English text. Additionally, Phoenix-7B is finetuned with multilingual instruction data while the other instruction-tuned models are mainly finetuned with English instruction data. In §6.3, we explore whether further finetuning the LLMs on our multilingual dialogue data leads to language generalization improvement.

**ChatGPT Performance** - Without finetuning, ChatGPT consistently excels over other LLMs in all languages, demonstrating its outstanding multilingual evaluation ability. This aligns with prior studies (Chen et al., 2023a). The superior performance is attributed to its instruction-tuning on a stronger foundation model, in terms of both parameters and pretraining data. It also benefits from higher quality instruction data for finetuning than models like Alpaca-7B and Phoenix-7B, and further gains from reinforcement learning from human feedback (RLHF), enhancing its alignment with human problem-solving skills.

## 6.2 Effects of Training on English Data Only

Table 4 shows that all models perform optimally on English evaluation datasets at both turn and dialogue levels, as expected. We can observe that PoE and FineD-Eval perform consistently well across languages. Interestingly, their performance matches their respective multilingual finetuned variants (Table 5), implying that XLM-R is a strong cross-lingual encoder.

Finetuning the LLMs on the English synthetic data not only brings improvements in English but also in other languages on turn-level datasets. The extent of improvements of different models can differ significantly across various languages. For example, the Alpaca-7B model, as shown in Table 4 and in the "LLMs-Zeroshot" section of Table 5, sees a more substantial performance improvement in Latin languages (>0.14), compared to Hindi and Arabic (<0.04). On the other hand, Phoenix-7B has a more consistent performance boost across different languages.

At the dialogue level, the average correlation score of LLaMA-2-7B after finetuning on the English synthetic data improves by over twice as much compared to when using zero-shot prompting (0.121 -> 0.292). For Baichuan-2-7B, the absolute improvement is around 5% (0.241 -> 0.293). However, the improvement brought by finetuning is less prominent for the instruction-tuned LLMs. For example, a slight improvement from 0.255 to 0.267

| Turn-Level | | | | | | | | | | | | |
|---|---|---|---|---|---|---|---|---|---|---|---|---|
| **Category** | **Models** | **EN** | **ZH** | **ES** | **DE** | **FR** | **JA** | **KO** | **HI** | **AR** | **RU** | **AVG** |
| BERT-Based | PoE† | 0.464 | 0.437 | 0.441 | 0.454 | 0.455 | 0.424 | 0.417 | 0.361 | 0.422 | 0.436 | 0.431 |
| LLMs-Zeroshot | LLaMA-7B | 0.038 | 0.025 | 0.094 | 0.028 | 0.037 | 0.071 | 0.015 | -0.020 | 0.016 | 0.072 | 0.038 |
| | LLaMA-2-7B | 0.065 | 0.076 | 0.084 | 0.029 | 0.033 | 0.101 | 0.108 | 0.066 | 0.073 | 0.010 | 0.064 |
| | BLOOM-7B | 0.044 | 0.134 | 0.100 | 0.019 | 0.084 | 0.017 | 0.005 | 0.048 | 0.099 | 0.062 | 0.061 |
| | Falcon-7B | 0.143 | 0.127 | 0.155 | 0.088 | 0.151 | 0.093 | 0.011 | 0.068 | 0.109 | 0.077 | 0.102 |
| | Baichuan-2-7B | 0.175 | 0.134 | 0.118 | 0.133 | 0.117 | 0.102 | 0.139 | 0.092 | 0.119 | 0.129 | 0.126 |
| | Alpaca-7B | 0.337 | 0.197 | 0.269 | 0.269 | 0.277 | 0.156 | 0.131 | 0.131 | 0.160 | 0.250 | 0.218 |
| | Vicuna-7B | 0.211 | 0.165 | 0.226 | 0.186 | 0.217 | 0.160 | 0.119 | 0.119 | 0.144 | 0.197 | 0.175 |
| | Phoenix-7B | 0.298 | 0.249 | 0.281 | 0.190 | 0.265 | 0.166 | 0.112 | 0.214 | 0.224 | 0.174 | 0.217 |
| | ChatGPT | 0.471 | 0.433 | 0.467 | 0.462 | 0.459 | 0.415 | 0.365 | 0.346 | 0.398 | 0.423 | 0.424 |
| LLMs-FT (ours) | LLaMA-7B† | 0.363 | 0.267 | 0.245 | 0.274 | 0.271 | 0.232 | 0.223 | 0.216 | 0.214 | 0.277 | 0.258 |
| | LLaMA-2-7B† | **0.565** | 0.484 | 0.510 | 0.506 | 0.523 | 0.436 | 0.416 | 0.355 | 0.378 | 0.478 | 0.465 |
| | BLOOM-7B† | 0.273 | 0.197 | 0.320 | 0.199 | 0.300 | 0.197 | 0.013 | 0.214 | 0.175 | 0.123 | 0.201 |
| | Falcon-7B† | 0.415 | 0.450 | 0.465 | 0.440 | 0.468 | 0.295 | 0.180 | 0.149 | 0.196 | 0.283 | 0.334 |
| | Baichuan-2-7B† | 0.541 | **0.505** | 0.515 | 0.501 | 0.513 | 0.453 | 0.444 | 0.388 | 0.412 | 0.480 | 0.475 |
| | Alpaca-7B† | 0.548 | 0.405 | 0.491 | 0.483 | 0.489 | 0.327 | 0.318 | 0.307 | 0.309 | 0.444 | 0.412 |
| | Phoenix-7B† | 0.481 | 0.435 | 0.461 | 0.366 | 0.465 | 0.323 | 0.264 | 0.410 | 0.435 | 0.334 | 0.397 |
| Ensemble (ours) | LLaMA-7B + PoE† | 0.476 | 0.443 | 0.448 | 0.462 | 0.466 | 0.431 | 0.423 | 0.371 | 0.425 | 0.442 | 0.439 |
| | LLaMA-2-7B + PoE † | 0.558 | 0.498 | **0.518** | **0.520** | **0.528** | **0.470** | **0.455** | 0.406 | 0.444 | **0.494** | **0.489** |
| | BLOOM-7B + PoE† | 0.485 | 0.444 | 0.461 | 0.460 | 0.474 | 0.425 | 0.418 | 0.376 | 0.431 | 0.440 | 0.441 |
| | Falcon-7B + PoE† | 0.494 | 0.479 | 0.485 | 0.488 | 0.499 | 0.419 | 0.400 | 0.355 | 0.411 | 0.437 | 0.447 |
| | Baichuan-2-7B + PoE† | 0.544 | 0.500 | 0.508 | 0.504 | 0.514 | 0.464 | **0.455** | 0.416 | 0.447 | 0.484 | 0.484 |
| | Alpaca-7B + PoE† | 0.543 | 0.461 | 0.503 | 0.504 | 0.511 | 0.420 | 0.412 | 0.387 | 0.413 | 0.476 | 0.463 |
| | Phoenix-7B + PoE† | 0.503 | 0.463 | 0.479 | 0.451 | 0.487 | 0.410 | 0.388 | **0.420** | **0.455** | 0.426 | 0.448 |
| Dialogue-Level | | | | | | | | | | | | |
| BERT-Based | FineD† | 0.386 | 0.354 | 0.362 | 0.362 | 0.372 | 0.346 | 0.341 | 0.343 | 0.339 | 0.376 | 0.358 |
| LLMs-Zeroshot | LLaMA-7B | 0.190 | 0.190 | 0.226 | 0.196 | 0.151 | 0.141 | 0.120 | 0.027 | 0.035 | 0.151 | 0.143 |
| | LLaMA-2-7B | 0.036 | 0.193 | 0.154 | 0.091 | 0.166 | 0.125 | 0.165 | 0.027 | 0.128 | 0.127 | 0.121 |
| | BLOOM-7B | 0.071 | 0.212 | 0.063 | 0.063 | 0.122 | 0.104 | 0.058 | 0.097 | 0.122 | 0.078 | 0.099 |
| | Falcon-7B | 0.286 | 0.240 | 0.248 | 0.268 | 0.153 | 0.113 | 0.107 | 0.134 | 0.168 | 0.219 | 0.194 |
| | Baichuan-2-7B | 0.296 | 0.316 | 0.270 | 0.258 | 0.274 | 0.211 | 0.198 | 0.156 | 0.201 | 0.235 | 0.241 |
| | Alpaca-7B | 0.441 | 0.321 | 0.386 | 0.404 | 0.402 | 0.301 | 0.268 | 0.208 | 0.270 | 0.356 | 0.336 |
| | Vicuna-7B | 0.347 | 0.234 | 0.243 | 0.260 | 0.242 | 0.209 | 0.220 | 0.132 | 0.148 | 0.231 | 0.226 |
| | Phoenix-7B | 0.312 | 0.292 | 0.264 | 0.261 | 0.291 | 0.254 | 0.163 | 0.253 | 0.253 | 0.206 | 0.255 |
| | ChatGPT | 0.419 | 0.375 | 0.407 | 0.395 | 0.404 | 0.378 | 0.310 | 0.324 | **0.385** | 0.363 | 0.376 |
| LLMs-FT (ours) | LLaMA-7B† | 0.237 | 0.201 | 0.192 | 0.208 | 0.240 | 0.173 | 0.169 | 0.151 | 0.172 | 0.207 | 0.195 |
| | LLaMA-2-7B† | 0.444 | 0.401 | 0.405 | 0.407 | 0.410 | 0.363 | 0.359 | 0.319 | 0.343 | 0.404 | 0.386 |
| | BLOOM-7B† | 0.289 | 0.235 | 0.269 | 0.249 | 0.253 | 0.175 | 0.132 | 0.288 | 0.274 | 0.136 | 0.230 |
| | Falcon-7B† | 0.376 | 0.366 | 0.314 | 0.334 | 0.320 | 0.231 | 0.146 | 0.142 | 0.197 | 0.174 | 0.260 |
| | Baichuan-2-7B† | 0.344 | 0.329 | 0.309 | 0.315 | 0.316 | 0.275 | 0.323 | 0.278 | 0.325 | 0.304 | 0.312 |
| | Alpaca-7B† | 0.420 | 0.362 | 0.383 | 0.394 | 0.379 | 0.309 | 0.263 | 0.255 | 0.278 | 0.351 | 0.339 |
| | Phoenix-7B† | 0.339 | 0.324 | 0.328 | 0.293 | 0.321 | 0.275 | 0.229 | 0.321 | 0.316 | 0.259 | 0.300 |
| Ensemble (ours) | LLaMA-7B + FineD† | 0.405 | 0.364 | 0.371 | 0.368 | 0.379 | 0.353 | 0.349 | 0.349 | 0.346 | 0.384 | 0.367 |
| | LLaMA-2-7B + FineD † | **0.477** | **0.434** | **0.434** | **0.436** | **0.442** | **0.399** | **0.394** | **0.380** | **0.385** | **0.438** | **0.422** |
| | BLOOM-7B + FineD† | 0.405 | 0.373 | 0.384 | 0.374 | 0.387 | 0.348 | 0.341 | 0.374 | 0.370 | 0.373 | 0.373 |
| | Falcon-7B + FineD† | 0.445 | 0.413 | 0.397 | 0.403 | 0.407 | 0.356 | 0.345 | 0.341 | 0.346 | 0.377 | 0.383 |
| | Baichuan-2-7B + FineD† | 0.402 | 0.379 | 0.366 | 0.371 | 0.374 | 0.339 | 0.369 | 0.333 | 0.369 | 0.364 | 0.367 |
| | Alpaca-7B + FineD† | 0.461 | 0.407 | 0.425 | 0.434 | 0.427 | 0.369 | 0.347 | 0.342 | 0.357 | 0.410 | 0.398 |
| | Phoenix-7B + FineD† | 0.403 | 0.373 | 0.377 | 0.356 | 0.379 | 0.340 | 0.317 | 0.368 | 0.363 | 0.338 | 0.361 |

Table 5: Language-wise average turn-level (over 12 datasets) and dialogue-level (over 6 datasets) Pearson correlations of different models. The Spearman results can be found in Table 17. "LLMs-Zeroshot" means models applied directly without finetuning, whereas "LLMs-FT" represents finetuned models. The best score for each language is highlighted in bold and models finetuned on synthetic dialogue data are accompanied with a †.

is observed for Phoenix-7B. The performance of Alpaca-7B even drops from 0.336 to 0.291. A possible explanation is that the instruction-tuned LLMs already possess certain knowledge necessary for multi-turn coherence evaluation. Finetuning on the synthetic data doesn't bring much additional knowledge.

## 6.3 Effects of Training on Multilingual Data

**BERT-Based Metrics** - We can observe from Table 5 that both PoE and FineD-Eval are strong metrics for multilingual dialogue evaluation. PoE achieves an average Pearson score of 0.431 at the turn level, outperforming ChatGPT and all the finetuned LLMs, except for LLaMA-2-7B and Baichuan-2-7B. FineD-Eval achieves 0.358 at the

dialogue level. Its performance is only slightly worse than ChatGPT zero-shot prompting and the finetuned LLaMA-2-7B. Second, the performance of both metrics is quite consistent across different languages. Hence, we can conclude that BERT-based discriminative metrics that work well on English dialogues can also generalize to other languages with strong multilingual encoders and multilingual training data.

**Finetuned LLMs** - Comparing "LLMs-FT" models with their "LLMs-Zeroshot" counterparts shows that finetuning the LLMs with multilingual synthetic dialogue data significantly improves both dialogue evaluation and language generalization. The observation confirms our claim in §1. For instance, LLaMA-7B zero-shot prompting, with an average Pearson correlation of 0.038, performs poorly in all languages. Finetuning LLaMA-7B on the multilingual synthetic data boosts the performance to 0.258 with a nearly 0.2 absolute improvement in most languages. At the dialogue level, LLaMA-7B's average Pearson correlation increases from 0.143 to 0.195. Similar observations can be made w.r.t. other "LLMs-FT" models. Notably, finetuning LLaMA-2-7B leads to the most significant improvement at both turn and dialogue levels, from 0.064 to 0.465 and 0.121 to 0.386 respectively.

When comparing Alpaca-7B to LLaMA-7B or Phoenix-7B to BLOOM-7B (in the "LLMs-FT" category), we can observe that the Alpaca-7B outperforms LLaMA-7B (or Phoenix-7B outperforms BLOOM-7B) by significant margins in all languages at both turn and dialogue levels. The observation suggests that a two-stage finetuning process, i.e., finetuning on general-purpose instruction data followed by finetuning on custom data, helps achieve stronger LLM-based dialogue evaluators.

Additionally, we can observe that the improvements of instruction-based LLMs (Alpaca-7B and Phoenix-7B) at the dialogue level are less significant than at the turn level. Such a finding is similar to what we have observed in §6.2. Future work may explore how to introduce high-quality data that carries richer information and benefits the coherence evaluation of multi-turn dialogues.

Lastly, while finetuning with multilingual data boosts LLMs' performance in all languages, variations still occur depending on their pretrained backbone model. For instance, Alpaca-7B, LLaMA-7B, and Falcon-7B, which use an English-focused pretrained backbone, perform optimally in Latin

languages. Meanwhile, Phoenix-7B and Baichuan-2-7B, with a multilingual pretrained backbone, display more consistent performance in different languages.

**Metric Ensemble** - The ensemble of the LLMs and the BERT-based metrics yields strong multilingual dialogue evaluators at both turn and dialogue levels. The best combinations, LLaMA-2-7B + PoE and LLaMA-2-7B + FineD-Eval outperform ChatGPT by 6.5% and 4.6% in terms of the average Pearson correlations at the turn and dialogue levels respectively, albeit without RLHF and the size of their trainable parameters is less than 7.5B. Finally, we can observe that ensemble generally leads to better multilingual dialogue evaluation capability than individual BERT-based or LLM-based metrics.

## 7 Conclusion

This paper introduces xDial-Eval, a multilingual dialogue evaluation benchmark featuring 14930 annotated turns and 8691 dialogues in 10 languages. Both automatic and human evaluation validate the high quality of xDial-Eval. Additionally, we examine the performance of BERT-based metrics and emerging LLMs on this benchmark. Key takeaways include: (1) SoTA BERT-based metrics, backed by strong cross-lingual encoders and multilingual training data, can effectively handle multilingual dialogue evaluation. (2) Recent general-purpose instruction-following LLMs show promise as unified multilingual dialogue evaluators. Future work could refine optimization techniques and use high-quality, task-specific data for fine-tuning the LLMs. (3) Ensembling BERT-based metrics and LLMs outperforms ChatGPT in correlation with human judgment and achieves good language generalization (4) Despite the ensemble approach's good correlations, multilingual automatic dialogue evaluation remains unsolved. xDial-Eval could serve as a valuable benchmark for tracking future research progress in this field.

## Limitations

Our investigations involving open-source LLMs are restricted to their 7B variants. Future research should explore whether improving the scale of these LLMs, such as using their respective 13B, 40B, or 65B variants, enhances their capability for zero-shot dialogue evaluation, and whether larger LLMs can better generalize to various languages after fine-tuning with multilingual dialogue data.

Additionally, open-domain dialogue evaluation is multi-faceted in nature, i.e., there are many evaluation aspects to consider at both turn and dialogue levels. Our research concentrates primarily on the context relevance of responses and the coherence of multi-turn dialogues respectively. Future studies might consider investigating how to prompt LLMs to assess various aspects, as well as exploring strategies to fine-tune these LLMs for optimization of multi-dimensional evaluation.

Furthermore, multilingual automatic dialogue evaluation, particularly at the dialogue level, remains an unsolved challenge. Despite the solid performance of BERT-based metrics and LLMs ensembles on xDial-Eval, there's considerable room for improvement. There are two major limitations:

(1) Subjectivity in dialogue evaluation - Determining a "good" or "successful" dialogue can be highly subjective, relying on factors such as the conversation's goals, participants, and social and cultural context. Sometimes, even human judges find it challenging (Smith et al., 2022). Possible solutions could involve generating expert-annotated dialogue data of high quality or creating custom metrics designed for varying evaluation scenarios.

(2) Long context modeling - Some dialogues in xDial-Eval can be quite lengthy, exceeding the maximum token limits of open-source LLMs (> 2048 tokens). We currently address this by truncation, which, unfortunately, results in information loss. Future research could focus on improving the modeling of long dialogues.

## Ethics Statement

The xDial-Eval benchmark originates from publicly accessible English datasets. To support and propel future exploration in the domain of automatic dialogue evaluation, and in harmony with the initiatives of other researchers, we commit to making the data and all our models open-source for future research endeavors.

## Acknowledgement

This work is supported by Human Robot Collaborative AI under its AME Programmatic Funding Scheme (Project No. A18A2b0046), the National Natural Science Foundation of China (Grant No. 62271432), and the Internal Project Fund from Shenzhen Research Institute of Big Data under Grant No. T00120220002. This work is also a result of the projects: ASTOUND (101071191 - HORIZON-EIC-2021-PATHFINDERCHALLENGES-01) funded by the European Commission, BE-WORD (PID2021-126061OB-C43) funded by MCIN/AEI/10.13039/501100011033 and, as appropriate, by "ERDF A way of making Europe", by the "European Union", and the Research Grants for Young Investigators from Universidad Politécnica de Madrid (GENIUS:APOYO-JOVENES-21-TAXTYC-32-K61X37) funded by Comunidad de Madrid. Finally, we also would like to thank Tencent AI Lab for providing support to this work.

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

## A  MT Evaluation Prompt Template

Table 6 is an example prompt we use for translation quality evaluation of xDial-Eval data with GPT-4.

## B  Examples of xDial-Eval Data

Table 7 and Table 8 show example data instances from the DailyDialog-Zhao (Zhao et al., 2020) turn-level and the IEval (Svikhnushina et al., 2022) dialogue-level datasets respectively.

## C  Model Descriptions

### C.1  BERT-Based Discriminative Metrics

**PoE** - consists of a pre-trained transformer encoder and a collection of domain-specific sub-metrics. Each sub-metric contains an adapter (Houlsby et al.,

============ PROMPT EXAMPLE ============

Given the following English source sentence, evaluate the adequacy of the French translation on a scale of 1 to 5. Adequacy is defined as "to what extent the translated sentence preserves the semantic meaning of the source sentence"

English source sentence:

I was not able to attend the lectures last week. Can you help me understand some concepts?

French translation:

Je n'ai pas pu assister aux conférences de la semaine dernière. Pouvez-vous m'aider à comprendre certains concepts?

Provide your output in the format:
adequacy: x
where x is the corresponding numerical rating.

Table 6: An example prompt template of machine translation evaluation with GPT-4

.

2019) and a scoring head. It is trained in a multi-task manner with large-scale multi-domain dialogue data. PoE scores a response by averaging the scores of all domain-specific sub-metrics.

**FineD-Eval** - targets multi-dimensional dialogue-level evaluation. It consists of three sub-metrics focusing on coherence, likability, and informativeness evaluation respectively. The sub-metrics are trained to rank the good-quality dialogue ahead of its poor-quality counterpart. FineD-Eval scores a dialogue by computing the arithmetic mean of individual sub-metric scores assigned to the dialogue.

### C.2  LLM-Based Generative Metrics

**LLaMA-Series** - LLaMA is a pre-trained causal LLM from Meta. It is currently the most popular open-source substitute for the closed-source GPT-3 (Brown et al., 2020). Its smallest variant, LLaMA-7B, is trained on 1 trillion tokens (mainly in English). LLaMA-2 is an optimized version of LLaMA with a better mixture of pre-training data and training strategies. LLaMA-2-7B is pretrained on 2T tokens, which is double that of LLaMA-7B. Alpaca is finetuned from LLaMA-7B on 52K instruction-following demonstrations. The data are generated from OpenAI's text-davinci-003 model with self-instruct (Wang et al., 2023). Vicuna is finetuned on 70K user-shared conversations collected from ShareGPT. Both models adopt the Low-Rank Adaptation (LoRA) (Hu et al., 2022) approach for finetuning. Both Alpaca and Vicuna

|  | **English** | **Chinese** | **Spanish** |
|---|---|---|---|
| **Context**: | well , how does it look ? 
 it 's a perfect fit . 
 let me pay for it now . | 嗯，看起来怎么样？ 
 这件非常合身。 
 让我现在付钱吧。 | bueno, ¿cómo se ve? 
 es un ajuste perfecto. 
 déjame pagarlo ahora. |
| **Response**: | cash , credit card , or debit card ? | 现金、信用卡还是借记卡？ | efectivo, tarjeta de crédito o tarjeta de débito? |
|  | **German** | **French** | **Japanese** |
| **Context**: | Nun, wie sieht es aus? 
 Es ist eine perfekte Passform. 
 Lass mich jetzt dafür bezahlen. | Eh bien, à quoi ça ressemble? 
 C'est un ajustement parfait. 
 Laissez-moi payer pour cela maintenant. | さて、それはどのようにえますか？ 
 それは完璧なフィット感です。 
 今それを支わせてください. |
| **Response**: | Bargeld, Kreditkarte oder Debitkarte? | Argent comptant, carte de crédit ou carte de débit? | 金、クレジットカド、またはデビットカド？ |
|  | **Korean** | **Russian** |  |
| **Context**: | 글쎄, 어떻게보이나요? 
 완벽한핏입니다. 
 지금지불하겠습니다. | ну, как это выглядит? 
 это идеально подходит. 
 позвольте мне заплатить за это сейчас. |  |
| **Response**: | 현금, 신용카드또는직불카드? | наличными, кредитной или дебетовой картой? |  |

Table 7: Example data instance from turn-level dataset - DailyDialog-Zhao (Zhao et al., 2020) in 8 different languages. Hindi and Arabic data are not included due to latex compatibility issues. The relevance rating of the response is 5 out of 5.

|  | **English** | **Chinese** |
|---|---|---|
| **Human**: | i was really glad i finished my service for the military | 我真的很高兴我完成了为军队的服务 |
| **Chatbot**: | What kind of service was it? | 这是什么样的服务？ |
| **Human**: | medical service | 医疗服务 |
| **Chatbot**: | Oh no, what was it for? | 哦不，这是干什么用的？ |
| **Human**: | For combat medical team | 对于战斗医疗队 |
| **Chatbot**: | What was it for? | 这是干什么用的？ |
|  | **Spanish** | **Japanese** |
| **Human**: | Estaba muy contento de haber terminado mi servicio para el ejército. | での奉仕をえて本当によかったです |
| **Chatbot**: | ¿Qué tipo de servicio era? | どんなサビスでしたか？ |
| **Human**: | Servicio Médico | 医サビス |
| **Chatbot**: | Oh no, ¿para qué fue? | ああ、いや、それは何のためでしたか？ |
| **Human**: | Para el equipo médico de combate | 医チム向け |
| **Chatbot**: | ¿Para qué sirvió? | それは何のためでしたか？ |
|  | **German** | **Korean** |
| **Human**: | Ich war wirklich froh, dass ich meinen Militärdienst beendet hatte | 군복무를마치게되어정말기뻤습니다 |
| **Chatbot**: | Um was für einen Service handelte es sich? | 어떤종류의봉사였습니까？ |
| **Human**: | Sanitätsdienst | 의료서비스 |
| **Chatbot**: | Oh nein, wozu war das gut? | 오, 아니, 그게뭐야？ |
| **Human**: | Für das medizinische Gefechtsteam | 전투의료팀을위해 |
| **Chatbot**: | Wozu diente es? | 그것은무엇을위한것인가？ |
|  | **French** | **Russian** |
| **Human**: | J'étais vraiment content d'avoir terminé mon service militaire | Я был очень рад, что закончил службу в армии |
| **Chatbot**: | De quel type de service s'agissait-il? | Что это была за услуга? |
| **Human**: | Service médical | медицинское обслуживание |
| **Chatbot**: | Oh non, à quoi servait-il? | О нет, для чего это было нужно? |
| **Human**: | Pour l'équipe médicale de combat | Для боевой медицинской команды |
| **Chatbot**: | À quoi servait-il? | Для чего это было нужно? |

Table 8: Example data instance from dialogue-level dataset - IEval (Svikhnushina et al., 2022) in 8 different languages. Hindi and Arabic data are not included due to latex compatibility issues. The overall rating of the dialogue is 1 out of 3.

are developed to mimic the general instruction-following capability of ChatGPT.

**BLOOM-Series** - BLOOM is a large-scale multilingual causal LLM released in the BigScience workshop. It is pretrained on the ROOTS corpus (Laurençon et al., 2022) in 46 natural languages and 13 programming languages. Phoenix is an open-source multilingual ChatGPT-like model that can handle 40+ languages. It is finetuned from BLOOMZ[14] on mixed multilingual instruction data and conversation data. The instruction data of Phoenix is collected with self-instruction (Wang et al., 2023) while the conversation data is mainly ChatGPT-distilled. As the collected data are mainly in English, they are then translated into other languages to support the post-training process of

---

[14]A variant of BLOOM finetuned on diverse multilingual data, which cover a wide array of NLP tasks, such as question answering, topic classification, and program synthesis.

| | |
|---|---|
| Human:
[H] My dog puked all over the carpet a few weeks back.
[C] Oh no! Did you clean it up?
[H] yes
[C] Did you clean it up?
[H] yes I did
[C] That's great! I'm glad it worked out for you!

Above is a dialogue between a human ([H]) and a chatbot ([C]). Is the dialogue coherent?

Assistant:
**No** | Human:
Dialogue Context:
hello , tell me something about yourself !
i am 56 years old and i am a housekeeper .
do you have any hobbies or interests ?

Dialogue Response:
i like to read and write poetry .

Given the dialogue context and response above, is the response relevant to the context?

Assistant:
**Yes** |
| (a) Dialogue-Level Evaluation | (b) Turn-Level Evaluation |

Table 9: Vicuna-7B Instruction Template.

| | |
|---|---|
| Human:\
[H] My dog puked all over the carpet a few weeks back.
[C] Oh no! Did you clean it up?
[H] yes
[C] Did you clean it up?
[H] yes I did
[C] That's great! I'm glad it worked out for you!

Above is a dialogue between a human ([H]) and a chatbot ([C]). Is the dialogue coherent?

\Assistant:\
**No** | Human:\
Dialogue Context:
hello , tell me something about yourself !
i am 56 years old and i am a housekeeper .
do you have any hobbies or interests ?

Dialogue Response:
i like to read and write poetry .

Given the dialogue context and response above, is the response relevant to the context?

\Assistant:\
**Yes** |
| (a) Dialogue-Level Evaluation | (b) Turn-Level Evaluation |

Table 10: Phoenix-7B Instruction Template.

| | |
|---|---|
| [H] My dog puked all over the carpet a few weeks back.
[C] Oh no! Did you clean it up?
[H] yes
[C] Did you clean it up?
[H] yes I did
[C] That's great! I'm glad it worked out for you!

Question: given the input dialogue between a human ([H]) and a chatbot ([C]), whether the dialogue is coherent?

Answer:
**No** | Dialogue Context:
hello , tell me something about yourself !
i am 56 years old and i am a housekeeper .
do you have any hobbies or interests ?

Dialogue Response:
i like to read and write poetry .

Question: given the context and response, predict whether the response is relevant to the context?

Answer:
**Yes** |
| (a) Dialogue-Level Evaluation | (b) Turn-Level Evaluation |

Table 11: Falcon-7B Instruction Template.

Phoenix.

**Falcon** - Falcon (Almazrouei et al., 2023) is a large-scale pretrained causal language model released by The Technology Innovation Institute. It is trained on 1,500B tokens of RefinedWeb (Penedo et al., 2023) enhanced with curated corpora, such as social media conversations, books, and technical papers. The training data is mainly in English and French.

**BaiChuan-2** - BaiChuan-2 (Yang et al., 2023) en-

 [INST] <<SYS>>
You are a helpful, respectful and honest assistant. Always answer as helpfully as possible, while being safe. Your answers should not include any harmful, unethical, racist, sexist, toxic, dangerous, or illegal content. Please ensure that your responses are socially unbiased and positive in nature.

If a question does not make any sense, or is not factually coherent, explain why instead of answering something not correct. If you don't know the answer to a question, please don't share false information.
<</SYS>>

Given the following dialogue between a human ([H]) and a chatbot ([C]), whether the dialogue is coherent

[H] My dog puked all over the carpet a few weeks back.
[C] Oh no! Did you clean it up?
[H] yes
[C] Did you clean it up?
[H] yes I did
[C] That's great! I'm glad it worked out for you!

[/INST]
**No**

(a) Dialogue-Level Evaluation

 [INST] <<SYS>>
You are a helpful, respectful and honest assistant. Always answer as helpfully as possible, while being safe. Your answers should not include any harmful, unethical, racist, sexist, toxic, dangerous, or illegal content. Please ensure that your responses are socially unbiased and positive in nature.

If a question does not make any sense, or is not factually coherent, explain why instead of answering something not correct. If you don't know the answer to a question, please don't share false information.
<</SYS>>

Given the context and response, predict whether the response is relevant to the context

Dialogue Context:
hello , tell me something about yourself !
i am 56 years old and i am a housekeeper .
do you have any hobbies or interests ?

Dialogue Response:
i like to read and write poetry .

[/INST]
**Yes**

(b) Turn-Level Evaluation

Table 12: LLaMA-2 Instruction Template.

<reserved_106>Given the input dialogue between a human ([H]) and a chatbot ([C]) below, whether the dialogue is coherent.

[H] My dog puked all over the carpet a few weeks back.
[C] Oh no! Did you clean it up?
[H] yes
[C] Did you clean it up?
[H] yes I did
[C] That's great! I'm glad it worked out for you!

<reserved_107>
**No**

(a) Dialogue-Level Evaluation

<reserved_106>Given the context and response below, predict whether the response is relevant to the context.

Dialogue Context:
hello , tell me something about yourself !
i am 56 years old and i am a housekeeper .
do you have any hobbies or interests ?

Dialogue Response:
i like to read and write poetry .

<reserved_107>
**Yes**

(b) Turn-Level Evaluation

Table 13: Baichuan-2 Instruction Template.

compasses a series of large-scale multilingual language models, consisting of models with 7 billion and 13 billion parameters, and is trained from scratch on a massive dataset comprising 2.6 trillion tokens. The pretraining data is collected from a variety of sources, such as general internet webpages, books, research papers, codebases, and more.

**ChatGPT** - is a closed-source general-purpose instruction-following conversational AI developed by OpenAI. The training of ChatGPT follows the instructGPT development pipeline (Ouyang and et al., 2022): (1) supervised finetuning from a strong pre-trained large language model on high-quality human-collected instruction data; (2) Align the finetuned language model with humans' intentions and goals leveraging reinforcement from hu-

man feedback. ChatGPT and its successor, GPT-4 (OpenAI, 2023) are currently the most powerful AI assistants that are highly capable of solving various NLP tasks. Many open-source LLMs, such as those described in the LLaMA-series and BLOOM-Series, are trained to mimic their abilities.

## D   More Examples on Instruction Template

This section presents the instruction templates of Vicuna-7B (Table 9), Phoenix-7B (Table 10), Falcon-7B (Table 11), LLaMA-2-7B (Table 12), Baichuan-2-7B (Table 13), and ChatGPT (Table 14). We try to closely follow the instruction templates of the open-source LLMs used during their supervised finetuning stage.

## E   Additional Analyses

Table 16 and Table 17 are the corresponding Spearman correlations of Table 4 and Table 5 respectively. Similar observations to the ones in §6 can be made with Table 16 and Table 17.

**Natural Non-English Dialogues** - In addition to the translation-based xDial-Eval benchmark, we analyze the performance of finetuned models on organic non-English dialogue data using five Chinese dialogue evaluation datasets[15] released by Rodríguez-Cantelar et al. (2023). Table 15 presents the detailed results. We can observe that Baichuan-2-7B is the best among all models at the turn level while LLaMA-2-7B performs the best at the dialogue level. Similar to the observations in §6.3, the ensemble of BERT-based metrics and LLMs leads to even stronger correlations with human evaluation. The results demonstrate that our proposed metrics are not just proficient in managing translation-based multilingual dialogue data, but they also deliver strong performance when applied to natural non-English data.

**Correlations Among Languages** - In this section, we analyze the interdependence of judgments given to dialogue data in different languages by different models. Specifically, we choose the FED-Turn (Mehri and Eskenazi, 2020a) and FED-Dial (Mehri and Eskenazi, 2020a) datasets for analysis. The purpose is to check whether the models can provide consistent judgments to multilingual dialogue data with the same semantic meanings.

---

[15] https://chateval.org/dstc11

Figure 1 and 2 show the inter-lingual Pearson correlations of FineD-Eval / PoE, ChatGPT, LLaMA-7B, Alpaca-7B, BLOOM-7B, and Phoenix-7B on FED-Dial and FED-Turn respectively.

We can observe that FineD-Eval and PoE both display very consistent inter-lingual correlation patterns. The correlations among language pairs are ~ 0.9 for FineD-Eval and > 0.7 for PoE. The observation showcases that both metrics can provide a consistent evaluation of translation-based multilingual dialogue data. Additionally, we can observe that ChatGPT displays stronger inter-lingual correlations on FED-Dial than on FED-Turn. The observation partially explains why ChatGPT has a more consistent performance across different languages at the dialogue level than at the turn level (refer to Table 5 and Table 17).

Furthermore, Phoenix-7B and Alpaca-7B exhibit stronger inter-lingual correlations than their respective backbone models, LLaMA-7B and BLOOM-7B. The observation supports our conclusion in §6.3 that a two-stage finetuning process (adapt instruction-tuned models on custom multilingual data) yields dialogue evaluators that are more robust across different languages.

Lastly, we observe that the inter-lingual correlations of Alpaca-7B are notably stronger within Latin-language pairs, as compared to other language combinations. In contrast, Phoenix-7B displays a more evenly distributed correlation pattern among various language pairs. This observation reinforces the findings presented in Section 6.3, suggesting that Alpaca-7B's performance excels in Latin languages, likely due to its English-centric pretrained backbone and initial fine-tuning on English instruction data. On the other hand, Phoenix, equipped with a multilingual pretrained backbone and an initial fine-tuning on multilingual instruction data, exhibits a more uniform performance across all languages.

**Correlations Among Metrics** - We delve deeper into examining the extent of agreement among the evaluations provided by various model-based metrics. To accomplish this, we calculate the correlations between different pairs of metrics, focusing specifically on their performance on turn-level datasets: FED-Turn & DailyDialog-Zhao (Zhao et al., 2020), and dialogue-level datasets: FED-Dial & IEval-Dial (Svikhnushina et al., 2022) in English. This approach allows us to discern whether these metrics complement each other. Figure 3

(a) Dialogue-Level Evaluation  (b) Turn-Level Evaluation

Table 14: ChatGPT Instruction Template.

| **Turn-Level** | | | | |
|---|---|---|---|---|
| **Models** | **ECM-Turn** | **LCCC-Turn** | **HC-Turn** | **Average** |
| PoE | 0.532 / 0.570 | 0.464 / 0.435 | 0.436 / 0.351 | 0.477 / 0.452 |
| Falcon-7B | 0.581 / 0.590 | 0.436 / 0.421 | 0.534 / 0.486 | 0.517 / 0.499 |
| Alpaca-7B | 0.476 / 0.480 | 0.409 / 0.390 | 0.482 / 0.447 | 0.456 / 0.439 |
| Phoenix-7B | 0.494 / 0.562 | 0.475 / 0.465 | 0.464 / 0.398 | 0.478 / 0.475 |
| LLaMA-2-7B | 0.527 / 0.551 | 0.436 / 0.415 | 0.515 / 0.471 | 0.493 / 0.479 |
| Baichuan-2-7B | 0.522 / 0.614 | 0.449 / 0.438 | 0.551 / 0.523 | 0.508 / 0.525 |
| PoE + Falcon-7B | 0.592 / 0.618 | 0.496 / 0.460 | 0.527 / 0.452 | 0.538 / 0.510 |
| PoE + Alpaca-7B | 0.555 / 0.573 | 0.481 / 0.446 | 0.503 / 0.441 | 0.513 / 0.487 |
| PoE + Phoenix-7B | 0.558 / 0.598 | 0.512 / 0.478 | 0.497 / 0.391 | 0.522 / 0.489 |
| PoE + LLaMA-2-7B | 0.576/ 0.607 | 0.494 / 0.457 | 0.528 / 0.462 | 0.532 / 0.508 |
| PoE + Baichuan-2-7B | 0.567 / 0.618 | 0.506 / 0.468 | 0.563 / 0.488 | 0.545 / 0.525 |
| **Dialogue-Level** | | | | |
| **Models** | **LCCC-Dial** | **HC-Dial** | **-** | **Average** |
| FineD | 0.386 / 0.379 | 0.563 / 0.578 | - | 0.475 / 0.479 |
| Falcon-7B | 0.329 / 0.338 | 0.678 / 0.698 | - | 0.504 / 0.518 |
| Alpaca-7B | 0.135 / 0.118 | 0.687 / 0.690 | - | 0.411 / 0.404 |
| Phoenix-7B | 0.245 / 0.234 | 0.692 / 0.696 | - | 0.469 / 0.465 |
| LLaMA-2-7B | 0.265 / 0.263 | 0.735 / 0.741 | - | 0.500 / 0.502 |
| Baichuan-2-7B | 0.252 / 0.253 | 0.675 / 0.706 | - | 0.464 / 0.480 |
| FineD + Falcon-7B | 0.456 / 0.456 | 0.712 / 0.727 | - | 0.584 / 0.592 |
| FineD + Alpaca-7B | 0.338 / 0.325 | 0.717 / 0.717 | - | 0.528 / 0.521 |
| FineD + Phoenix-7B | 0.353 / 0.349 | 0.722 / 0.722 | - | 0.538 / 0.536 |
| FineD + LLaMA-2-7B | 0.403 / 0.401 | 0.751 / 0.758 | - | 0.577 / 0.580 |
| FineD + Baichuan-2-7B | 0.345 / 0.342 | 0.709 / 0.719 | - | 0.527 / 0.531 |

Table 15: Pearson / Spearman correlations of different models (after finetuning on the multilingual dialogue data) on natural Chinese dialogue evaluation datasets.

presents the inter-metric Pearson correlations. We can observe that the judgments provided by different metrics are not consistent and the correlation patterns differ on different datasets. Furthermore, while the correlations between PoE and Alpaca-7B, and between PoE and Phoenix-7B, are reasonably good, they aren't excessively strong. This suggests that these pairs of metrics complement each other without being overly similar. This insight hints at the potent performance of an ensemble comprising PoE and Alpaca-7B, or PoE and Phoenix-7B, particularly on turn-level datasets. Similar observations can be made w.r.t. FineD-Eval and Alpaca-7B and FineD-Eval and Phoenix-7B. These insights provide a rationale for the conclusions we drew in the "Metric Ensemble" subsection of § 6.3, where we found that an ensemble of BERT-based metrics and LLMs creates highly effective multilingual

| Turn-Level | | | | | | | | | | | |
|---|---|---|---|---|---|---|---|---|---|---|---|
| **Models** | **EN** | **ZH** | **ES** | **DE** | **FR** | **JA** | **KO** | **HI** | **AR** | **RU** | **AVG** |
| PoE | 0.465 | 0.431 | 0.438 | 0.442 | 0.444 | 0.419 | 0.411 | 0.349 | 0.409 | 0.426 | 0.423 |
| Alpaca-7B | 0.482 | 0.336 | 0.405 | 0.419 | 0.418 | 0.248 | 0.029 | 0.155 | 0.186 | 0.346 | 0.302 |
| Phoenix-7B | 0.456 | 0.409 | 0.399 | 0.325 | 0.434 | 0.288 | 0.202 | 0.386 | 0.409 | 0.269 | 0.358 |
| LLaMA-2-7B | 0.523 | 0.401 | 0.456 | 0.415 | 0.450 | 0.324 | 0.324 | 0.242 | 0.248 | 0.388 | 0.377 |
| Baichuan-2-7B | 0.561 | 0.556 | 0.476 | 0.473 | 0.481 | 0.382 | 0.349 | 0.254 | 0.307 | 0.417 | 0.426 |
| **Dialogue-Level** | | | | | | | | | | | |
| FineD | 0.376 | 0.346 | 0.356 | 0.345 | 0.351 | 0.362 | 0.345 | 0.301 | 0.299 | 0.342 | 0.342 |
| Alpaca-7B | 0.400 | 0.306 | 0.351 | 0.338 | 0.361 | 0.205 | 0.208 | 0.189 | 0.172 | 0.309 | 0.284 |
| Phoenix-7B | 0.354 | 0.346 | 0.267 | 0.242 | 0.317 | 0.245 | 0.175 | 0.272 | 0.328 | 0.206 | 0.275 |
| LLaMA-2-7B | 0.375 | 0.288 | 0.334 | 0.347 | 0.322 | 0.246 | 0.244 | 0.199 | 0.209 | 0.304 | 0.287 |
| Baichuan-2-7B | 0.371 | 0.338 | 0.270 | 0.311 | 0.317 | 0.271 | 0.278 | 0.224 | 0.249 | 0.277 | 0.291 |

Table 16: Language-wise average turn-level (over 12 datasets) and dialogue-level (over 6 datasets) Spearman correlations of models finetuned on English data only.

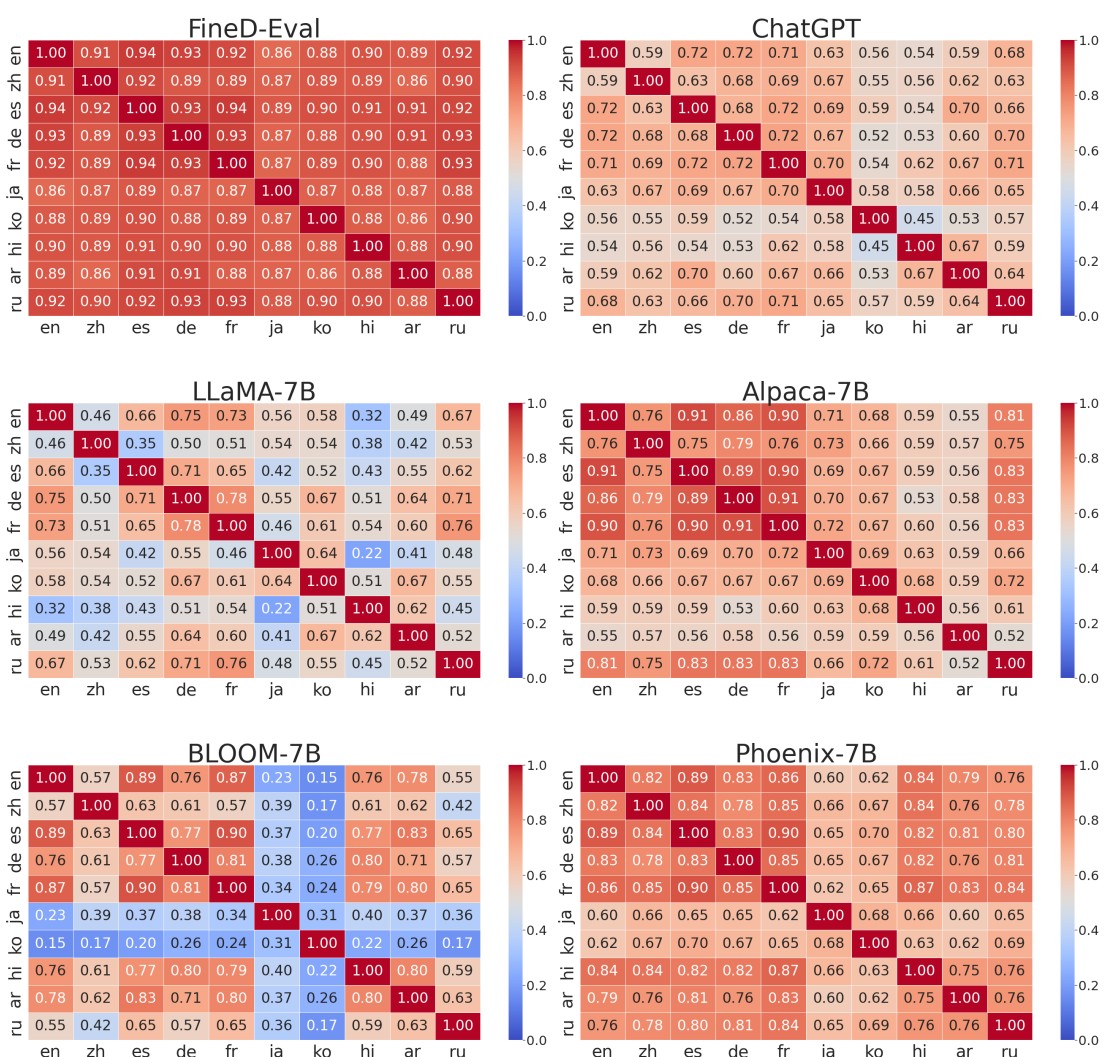

Figure 1: Inter-lingual Pearson correlations of different models on FED-Dial dataset.

dialogue evaluators that outperform ChatGPT.

## F Reproducibility

**Computation Details** - All experiments were carried out using a single 40GB A100 GPU card. We

| Turn-Level | | | | | | | | | | | | |
|---|---|---|---|---|---|---|---|---|---|---|---|---|
| Category | Models | EN | ZH | ES | DE | FR | JA | KO | HI | AR | RU | AVG |
| BERT-Based | PoE† | 0.474 | 0.440 | 0.452 | 0.455 | 0.459 | 0.422 | 0.423 | 0.367 | 0.424 | 0.438 | 0.435 |
| LLMs-Zeroshot | LLaMA-7B | 0.038 | 0.034 | 0.083 | 0.028 | 0.038 | 0.058 | 0.014 | -0.009 | 0.015 | 0.059 | 0.036 |
| | LLaMA-2-7B | 0.068 | 0.081 | 0.083 | 0.041 | 0.041 | 0.101 | 0.108 | 0.070 | 0.073 | 0.022 | 0.069 |
| | BLOOM-7B | 0.046 | 0.144 | 0.101 | 0.010 | 0.080 | 0.023 | 0.002 | 0.043 | 0.088 | 0.072 | 0.061 |
| | Falcon-7B | 0.144 | 0.121 | 0.154 | 0.094 | 0.141 | 0.093 | 0.011 | 0.069 | 0.108 | 0.076 | 0.101 |
| | Baichuan-2-7B | 0.178 | 0.156 | 0.128 | 0.138 | 0.132 | 0.107 | 0.153 | 0.097 | 0.135 | 0.134 | 0.136 |
| | Alpaca-7B | 0.340 | 0.202 | 0.266 | 0.281 | 0.278 | 0.157 | 0.136 | 0.129 | 0.161 | 0.244 | 0.219 |
| | Vicuna-7B | 0.205 | 0.162 | 0.225 | 0.178 | 0.209 | 0.166 | 0.114 | 0.113 | 0.144 | 0.202 | 0.172 |
| | Phoenix-7B | 0.300 | 0.283 | 0.277 | 0.188 | 0.284 | 0.188 | 0.136 | 0.229 | 0.270 | 0.184 | 0.234 |
| | ChatGPT | 0.479 | 0.435 | 0.473 | 0.474 | 0.463 | 0.414 | 0.364 | 0.347 | 0.397 | 0.430 | 0.428 |
| LLMs-FT (ours) | LLaMA-7B† | 0.363 | 0.268 | 0.238 | 0.261 | 0.261 | 0.226 | 0.222 | 0.216 | 0.208 | 0.266 | 0.253 |
| | LLaMA-2-7B† | 0.553 | 0.470 | 0.501 | 0.500 | 0.516 | 0.424 | 0.409 | 0.350 | 0.375 | 0.468 | 0.457 |
| | BLOOM-7B† | 0.315 | 0.233 | 0.359 | 0.229 | 0.321 | 0.219 | -0.039 | 0.232 | 0.184 | 0.138 | 0.219 |
| | Falcon-7B† | 0.423 | 0.439 | 0.466 | 0.449 | 0.479 | 0.282 | 0.166 | 0.146 | 0.185 | 0.275 | 0.331 |
| | Baichuan-2-7B† | **0.569** | **0.514** | **0.523** | 0.509 | 0.517 | **0.461** | 0.446 | 0.388 | 0.415 | 0.478 | **0.482** |
| | Alpaca-7B† | 0.561 | 0.397 | 0.493 | 0.483 | 0.486 | 0.328 | 0.310 | 0.300 | 0.304 | 0.437 | 0.410 |
| | Phoenix-7B† | 0.491 | 0.428 | 0.467 | 0.359 | 0.469 | 0.311 | 0.252 | **0.421** | 0.431 | 0.327 | 0.396 |
| Ensemble (ours) | LLaMA-7B + PoE† | 0.482 | 0.437 | 0.444 | 0.456 | 0.462 | 0.426 | 0.427 | 0.368 | 0.420 | **0.434** | 0.439 |
| | LLaMA-2-7B + PoE† | 0.552 | 0.488 | 0.510 | **0.512** | **0.520** | 0.457 | 0.445 | 0.390 | 0.432 | 0.483 | 0.479 |
| | BLOOM-7B + PoE† | 0.498 | 0.450 | 0.472 | 0.456 | 0.477 | 0.418 | 0.423 | 0.379 | 0.436 | 0.446 | 0.445 |
| | Falcon-7B + PoE† | 0.495 | 0.467 | 0.477 | 0.478 | 0.494 | 0.385 | 0.367 | 0.328 | 0.379 | 0.408 | 0.428 |
| | Baichuan-2-7B + PoE† | 0.553 | 0.503 | 0.513 | 0.506 | 0.514 | **0.461** | **0.454** | 0.401 | 0.440 | **0.480** | **0.482** |
| | Alpaca-7B + PoE† | 0.555 | 0.451 | 0.510 | 0.507 | 0.513 | 0.407 | 0.396 | 0.369 | 0.394 | 0.396 | 0.450 |
| | Phoenix-7B + PoE† | 0.506 | 0.453 | 0.481 | 0.433 | 0.485 | 0.385 | 0.365 | 0.413 | 0.448 | 0.406 | 0.438 |
| Dialogue-Level | | | | | | | | | | | | |
| BERT-Based | FineD† | 0.379 | 0.347 | 0.359 | 0.355 | 0.364 | 0.343 | 0.338 | 0.331 | 0.331 | 0.371 | 0.352 |
| LLMs-Zeroshot | LLaMA-7B | 0.174 | 0.172 | 0.205 | 0.181 | 0.130 | 0.155 | 0.128 | 0.012 | 0.022 | 0.131 | 0.131 |
| | LLaMA-2-7B | 0.037 | 0.190 | 0.156 | 0.104 | 0.166 | 0.123 | 0.182 | 0.030 | 0.146 | 0.125 | 0.126 |
| | BLOOM-7B | 0.086 | 0.208 | 0.074 | 0.076 | 0.145 | 0.125 | 0.072 | 0.098 | 0.133 | 0.097 | 0.111 |
| | Falcon-7B | 0.272 | 0.236 | 0.233 | 0.254 | 0.117 | 0.083 | 0.133 | 0.145 | 0.169 | 0.221 | 0.186 |
| | Alpaca-7B | 0.436 | 0.324 | 0.387 | 0.397 | 0.389 | 0.294 | 0.271 | 0.211 | 0.273 | 0.345 | 0.333 |
| | Baichuan-2-7B | 0.299 | 0.293 | 0.261 | 0.258 | 0.251 | 0.205 | 0.208 | 0.128 | 0.193 | 0.214 | 0.231 |
| | Vicuna-7B | 0.314 | 0.229 | 0.250 | 0.253 | 0.227 | 0.210 | 0.225 | 0.145 | 0.160 | 0.220 | 0.223 |
| | Phoenix-7B | 0.278 | 0.288 | 0.228 | 0.234 | 0.251 | 0.254 | 0.158 | 0.237 | 0.239 | 0.191 | 0.236 |
| | ChatGPT | 0.406 | 0.362 | 0.394 | 0.388 | 0.394 | 0.368 | 0.302 | 0.313 | **0.378** | 0.355 | 0.366 |
| LLMs-FT (ours) | LLaMA-7B† | 0.228 | 0.194 | 0.203 | 0.226 | 0.257 | 0.181 | 0.160 | 0.154 | 0.179 | 0.215 | 0.200 |
| | LLaMA-2-7B† | 0.228 | 0.194 | 0.203 | 0.226 | 0.257 | 0.181 | 0.160 | 0.154 | 0.179 | 0.215 | 0.200 |
| | BLOOM-7B† | 0.315 | 0.263 | 0.268 | 0.246 | 0.271 | 0.159 | 0.134 | 0.283 | 0.279 | 0.129 | 0.235 |
| | Falcon-7B† | 0.353 | 0.349 | 0.300 | 0.314 | 0.312 | 0.205 | 0.123 | 0.140 | 0.184 | 0.145 | 0.243 |
| | Baichuan-2-7B† | 0.365 | 0.337 | 0.330 | 0.327 | 0.335 | 0.298 | 0.342 | 0.279 | 0.325 | 0.327 | 0.327 |
| | Alpaca-7B† | 0.393 | 0.343 | 0.360 | 0.370 | 0.354 | 0.292 | 0.249 | 0.246 | 0.261 | 0.328 | 0.320 |
| | Phoenix-7B† | 0.346 | 0.341 | 0.338 | 0.303 | 0.325 | 0.274 | 0.231 | 0.330 | 0.321 | 0.259 | 0.307 |
| Ensemble (ours) | LLaMA-7B + FineD† | 0.395 | 0.358 | 0.371 | 0.361 | 0.373 | 0.353 | 0.346 | 0.343 | 0.341 | 0.381 | 0.362 |
| | LLaMA-2-7B + FineD† | **0.457** | **0.418** | **0.431** | **0.427** | **0.427** | 0.387 | **0.383** | 0.370 | 0.375 | **0.429** | **0.410** |
| | BLOOM-7B + FineD† | 0.398 | 0.364 | 0.378 | 0.362 | 0.379 | 0.346 | 0.338 | 0.363 | 0.364 | 0.368 | 0.366 |
| | Falcon-7B + FineD† | 0.430 | 0.403 | 0.393 | 0.393 | 0.397 | 0.344 | 0.337 | 0.326 | 0.337 | 0.365 | 0.372 |
| | Baichuan-2-7B + FineD† | 0.407 | 0.381 | 0.372 | 0.375 | 0.380 | 0.349 | 0.379 | 0.330 | 0.364 | 0.376 | 0.371 |
| | Alpaca-7B + FineD† | 0.441 | 0.390 | 0.411 | 0.417 | 0.409 | 0.357 | 0.338 | 0.330 | 0.342 | 0.397 | 0.383 |
| | Phoenix-7B + FineD† | 0.402 | 0.377 | 0.386 | 0.359 | 0.382 | 0.341 | 0.320 | **0.373** | 0.363 | 0.338 | 0.364 |

Table 17: Language-wise average turn-level (over 12 datasets) and dialogue-level (over 6 datasets) Spearman correlations of different models. "LLMs-Zeroshot" means models applied directly without finetuning, whereas "LLMs-FT" represents finetuned models. The best score for each language is highlighted in bold and models finetuned on synthetic dialogue data are accompanied with a †.

use the alpaca-lora library[16] for LLM fine-tuning with low-rank adaptation. During finetuning, we utilize a batch size of 128 with 4 gradient accumulation steps. The learning rate, cutoff length, dropout rate, and number of epochs are set to 0.0003, 1024, 0.05, and 3 respectively. For the LoRA parameters, the rank ($r$) and alpha are set to 8 and 16 correspondingly. The target module for low-rank adaptation includes the query and key projection matrices. The total number of trainable parameters is approximately 4M, which accounts for around 5.5% to 6% of the LLMs' full parameter size. The finetuning of each LLM with LoRA takes around 100 hours on 200K data. For training PoE and FineD-Eval, we follow the training procedures and hyperparameter settings introduced in their respective papers.

---

[16] https://github.com/tloen/alpaca-lora

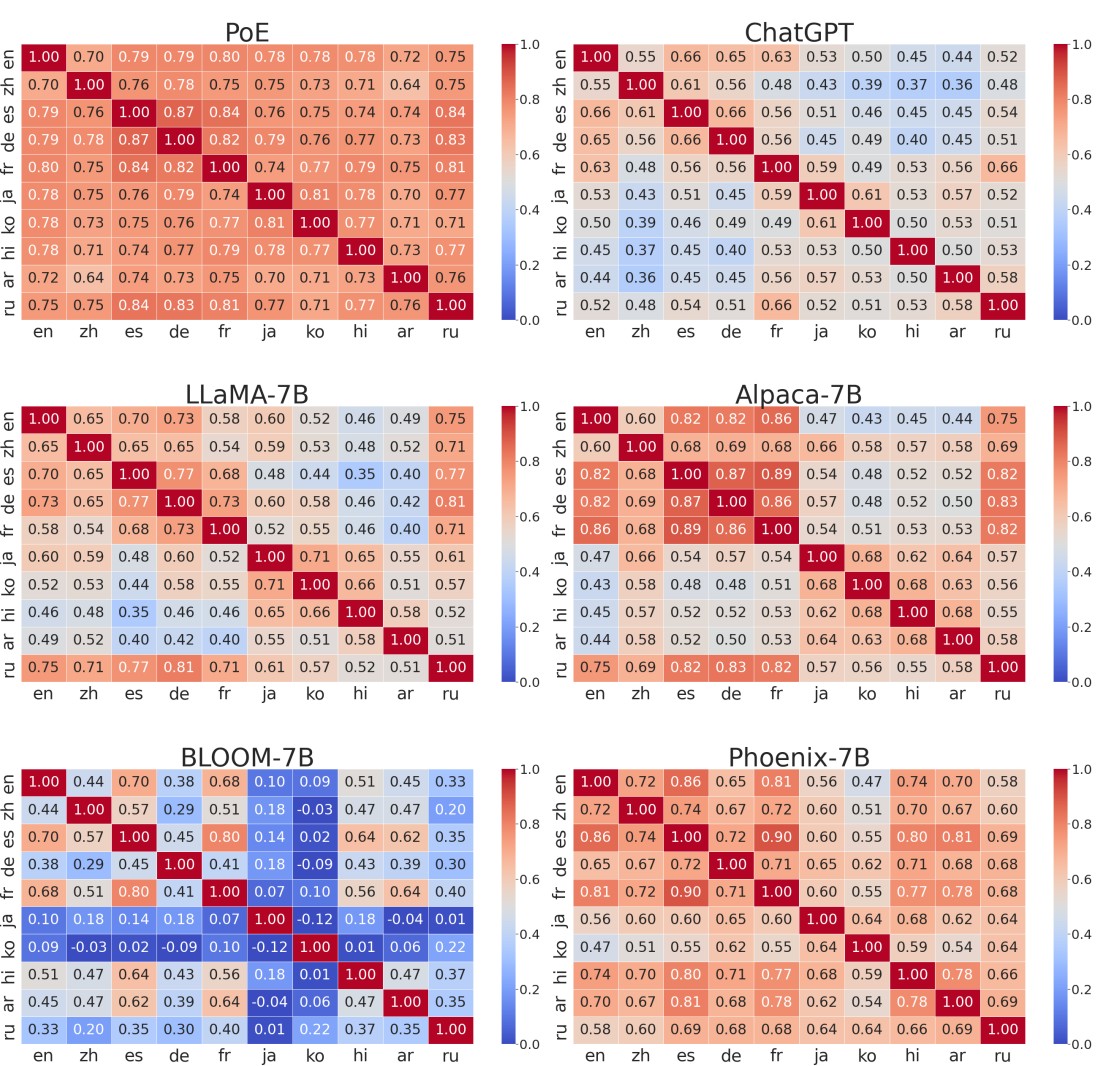

Figure 2: Inter-lingual Pearson correlations of different models on FED-Turn dataset.

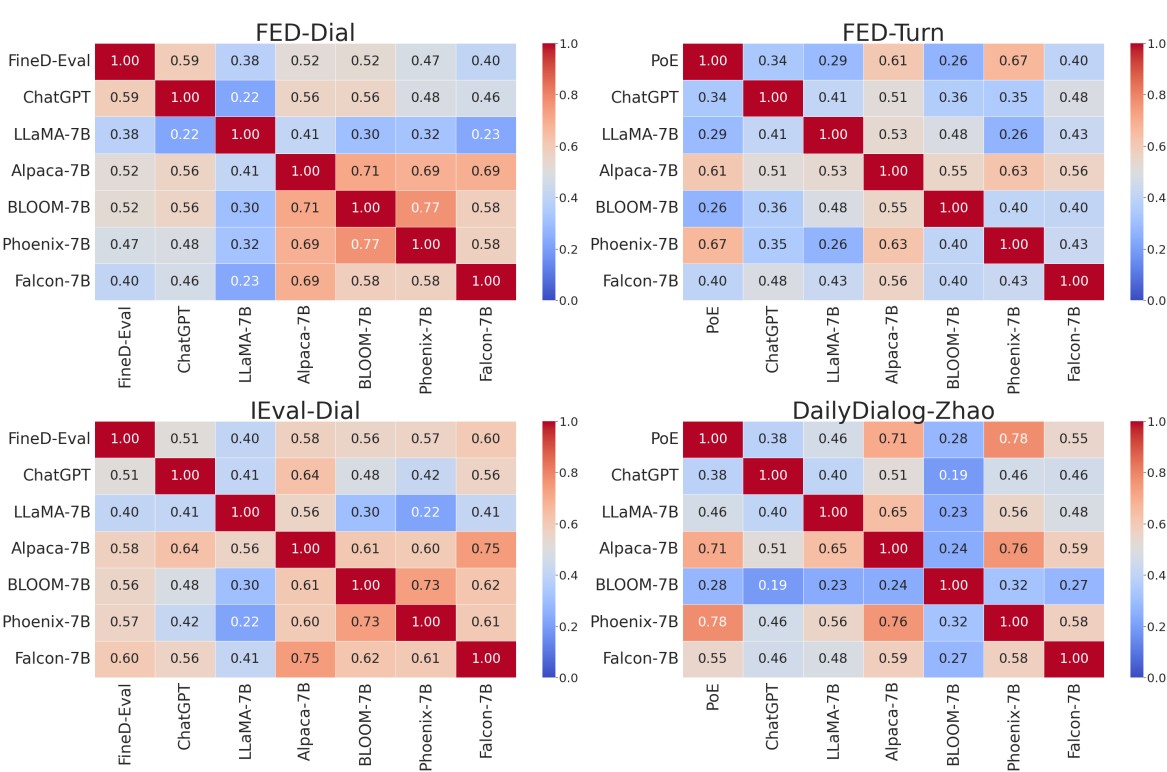

Figure 3: Inter-Metric Pearson correlations on different datasets.