# OpenReview forum: "xDial-Eval: A Multilingual Open-Domain Dialogue Evaluation Benchmark"
_EMNLP/2023/Conference — EMNLP 2023 Findings_

### Official Review · Reviewer_8yTD · 2023-08-01

**Soundness:** 3

**Excitement:**

3: Ambivalent: It has merits (e.g., it reports state-of-the-art results, the idea is nice), but there are key weaknesses (e.g., it describes incremental work), and it can significantly benefit from another round of revision. However, I won't object to accepting it if my co-reviewers champion it.

**Paper Topic And Main Contributions:**

- What is this paper about? This paper aims to provide a new multilingual open-domain dialogue evaluation benchmark including datasets, evaluation methods, and current metrics analysis.
- what contributions does it make? This paper collects 9 languages versions of multiple dialogue datasets with machine translation, analyzes current reference-free metric performance on the proposed datasets, and proposes new metrics.

**Questions For The Authors:**

1. How long will it cost to run the whole evaluation of the proposed benchmark with the Ensemble metric such as LLama-7B on a typical GPU?

**Reasons To Accept:**

- This paper constructs xDial-Eval which contains 12 turn-level and 6 dialogue-level open-source datasets. The original English version datasets are translated into nine different languages with commercial machine translation models.
- This paper assesses current discriminative and generative reference-free metrics on the proposed multilingual benchmark. The most recent LLMs are also evaluated in this paper.
- This paper introduces an ensemble metric for the proposed multilingual benchmark by combining generative and discrimination models.

**Reasons To Reject:**

- One of the concerns about this work is that the impact of the machine translation models on multiple parts. The datasets are translated and the synthetic datasets for fine-tuning are translated. If some sampled instances from the benchmark can be evaluated by human, it could be better.
- Regarding the proposed metric which combines LLMs and FineD, one concern is the computational limitations of the open-source LLMs and the training of the BERT-style model, especially for 9 languages of 18 datasets.

**Reproducibility:**

3: Could reproduce the results with some difficulty. The settings of parameters are underspecified or subjectively determined; the training/evaluation data are not widely available.

**Reviewer Confidence:**

3: Pretty sure, but there's a chance I missed something. Although I have a good feel for this area in general, I did not carefully check the paper's details, e.g., the math, experimental design, or novelty.

---

> ### Author Rebuttal · Authors · 2023-08-27
>
> **RQ1**. “One of the concerns about this work is that the impact of the machine translation models on multiple parts. The datasets are translated and the synthetic datasets for fine-tuning are translated. If some sampled instances from the benchmark can be evaluated by human, it could be better.”
>
> - We want to clarify that the **impact of machine translation quality is not significant and it is not the main concern of the paper**, because we are targeting evaluating coherence and contextual relevance in multilingual dialogues. If the reviewer checks the correlation results in Table 4 and Table 5, we can observe that **even just finetuning the models on the original English synthetic data , which are free from machine translation errors, brings improvements in different languages**. This suggests that the finetuning does help the models to capture high-level dialogue representation rather than surface-level syntax.
>
> - Additionally, **we are using state-of-the-art commercial translation service to translate the English evaluation data, the quality of the translation is very good as indicated by multiple automatic evaluation measures**. As shown in the table below, the score range of GPT-4 is from 1 to 5, that of BLEU is from 0 to 100, and that of WMT22-cometkiwi-DA, BLEURT, and BERTScore is from 0 to 1. We can see that all automatic metrics’ scores are high.
>
>     | Metrics | EN-Zh | EN-ES | EN-DE | EN-FR | EN-JA | EN-KO | EN-HI | EN-AR | EN-RU |
>     |-----------|-----------|-----------|-----------|-----------|-----------|-----------|-----------|-----------|-----------|
>     | GPT-4 ↑ | 4.58 | 4.64 | 4.63 | 4.59 | 4.33 | 4.26 | 4.40 | 4.39 | 4.40 |
>     | WMT22-cometkiwi-DA ↑ | 0.827 | 0.845 | 0.834 | 0.842 | 0.854  | 0.846 | 0.793 | 0.835 | 0.825 |
>     | BLEU (with back-translation) ↑ | 37.85 | 56.58 | 50.30 | 52.10 | 42.70 | 40.23 | 49.33 | 48.54 | 49.13 |
>     | BLEURT (with back-translation) ↑ | 0.773 | 0.820 | 0.817 | 0.811 | 0.786 | 0.782 | 0.797 | 0.794 | 0.804 |
>     | BERTScore (with back-translation) ↑ | 0.691 | 0.767 | 0.754 | 0.748 | 0.711 | 0.707 | 0.726 | 0.737 | 0.741 |
>
> - **For the reviewer’s reference, we have conducted additional human evaluation**. We collaborated with a service provider to have native speakers, proficient in English and their native language, evaluate the quality of translations from English to their mother tongue. Each reviewer assessed 350 translation pairs, rating them on a 1-5 scale, where 1 is poor and 5 is excellent. These evaluations covered 9 language pairs, totalling 3150 instances. A quality check was done on a random subset to ensure reliability. For the English-to-Chinese translations, three experts evaluated them. The inter-annotator agreement was 0.578, indicating medium agreement among experts. **The human evaluation results below agree with the automatic measures that the translation quality of the evaluation data is good**.
>
>     | Metrics | EN-Zh | EN-ES | EN-DE | EN-FR | EN-JA | EN-KO | EN-HI | EN-AR | EN-RU |
>     |-----------|-----------|-----------|-----------|-----------|-----------|-----------|-----------|-----------|-----------|
>     | Human ↑ | 4.75 | 4.68 | 4.59 | 4.53 | 4.39  | 4.32 | 4.17 | 4.41 | 4.71 |
>
> - Lastly, it is true that high-quality translation helps better finetuning of the models. However, even finetuning on multilingual synthetic data, whose translation quality is worse than that of the evaluation data, we can still observe significant correlation improvements in table 5 across different languages. **This observation reinforces the idea that the impact of machine translation is not significant on multilingual dialogue evaluation**.
>
> **RQ2**. “Regarding the proposed metric which combines LLMs and FineD, one concern is the computational limitations of the open-source LLMs and the training of the BERT-style model, especially for 9 languages of 18 datasets.”
>
> - We want to clarify that **the models are not trained on the 18 evaluation datasets**. We are finetuning the models on a synthetic multilingual dialogue dataset, which is automatically constructed. Hence, our methods are based on self-supervised learning. After model finetuning, the same models will be applied to evaluate the 18 datasets spanning acorss 10 different languages.
>
> - As stated in our paper, the total size of the synthetic training data is 200K. The finetuning of RoBERTa model can be easily completed with a single NVIDIA 3090TI 24GB GPU while the low-rank adaptation (LoRA) finetuning of a LLaMA-7B model can be completed with a single NVIDIA 4090TI 40GB GPU. Additionally, as stated in section F of the supplementary, the time taken for LoRA finetuning of a 7B LLM takes roughly 100 GPU hours while the finetuning of RoBERTa is much faster, typically, it takes less than 24 GPU hours. **Hence, the computation costs are not expensive**.
>
> **RQ3**. “How long will it cost to run the whole evaluation of the proposed benchmark with the Ensemble metric such as LLama-7B on a typical GPU?”
>
> - We have 14930 x 10 = 149300 annotated turns and 8691 * 10 = 86910 annotated multi-turn dialogues in the benchmark. When inference with LLaMA-7B on a single NVIDIA 3090TI 24GB GPU, we set the batch size to 1.
>
> - For turn-level evaluation,  the inference speed of LLaMA-7B is 4.65 instances per second on average. Hence, it takes around $149300 / 4.65 / 3600  \approx 8.92$ GPU hours to complete. For dialogue-level evaluation, the inference speed of LLaMA-7B is 3.17 instances per second on average. Note that the speed is slower than that of turn-level evaluation due to the longer dialogue contexts to process for multi-turn dialogues. Hence, it takes around $ 86910 / 3.17 / 3600 \approx 7.62 $ GPU hours to complete. In total, it costs $ 8.92 + 7.62 = 16.54 $ GPU hours to finish inference on the entire benchmark.
>
> - For the RoBERT-based metric, the inference speed is much faster due to its much smaller parameter size and larger inference batch size. For example, if we use a batch size of 16, the inference speed for dialogue-level evaluation is 20.8 instances per second and it takes $  86910 / 20.8 / 3600 \approx 1.16 $ GPU hours to complete the entire dialogue level evaluation. For turn-level evaluation, the inference speed is 44.8 instances per second and it takes $  149300 / 44.8 / 3600 \approx 0.93 $ GPU hours to complete the entire turn level evaluation. The total cost will be $ 1.16 + 0.93 = 2.09 $ GPU hours to finish inference on the entire benchmark.
>
> - **In summary, the costs of ensemble the two metrics are much lower compared to inference with commercial LLMs, such as ChatGPT, in terms of both money and inference time**.

---

### Official Review · Reviewer_SAJN · 2023-08-05

**Soundness:** 3

**Excitement:**

3: Ambivalent: It has merits (e.g., it reports state-of-the-art results, the idea is nice), but there are key weaknesses (e.g., it describes incremental work), and it can significantly benefit from another round of revision. However, I won't object to accepting it if my co-reviewers champion it.

**Paper Topic And Main Contributions:**

This work proposes xDial-Eval, an evaluation benchmark for multilingual open domain dialogue by machine translating 18 English open domain dialogue datasets into 8 languages. To demonstrate the usability of the dataset, several metrics BERT-based/generation-based LLMs are evaluated on the benchmark. The work also suggests ensembling metrics to be better than commercial LLM systems

The contribution is primarily a resource contribution followed by NLP engineering experiment.

**Questions For The Authors:**

A. How are the human annotations for evaluation obtained? What are the scores from the metrics compared against to compute the correlation?
(After the rebuttal, please also write this clearly in the experimentation section)

**Reasons To Accept:**

Good choice of languages for the dataset.

Good number and types of LLMs used for comparison.

Open domain multilingual conversations have received less interest and this work is a contribution towards that direction

**Reasons To Reject:**

The proposed dataset is an aggregation of different dialogue datasets focussing on different properties of dialogue - yet the emphasis is more on “coherence” and excludes other properties of dialgoue evaluation.

Further, the proposed dataset, despite being an evaluation benchmark for multilingual dialogue, uses automatic translation APIs and quality evaluation is done only on automatic metrics.

**Reproducibility:**

2: Would be hard pressed to reproduce the results. The contribution depends on data that are simply not available outside the author's institution or consortium; not enough details are provided.

**Reviewer Confidence:**

4: Quite sure. I tried to check the important points carefully. It's unlikely, though conceivable, that I missed something that should affect my ratings.

**Typos Grammar Style And Presentation Improvements:**

A. It is disputed whether LLMs can be used for translation quality evaluation as zero-shot , further a lot of segment level evaluation metrics are not yet equipped to do quality evaluation(https://aclanthology.org/2023.acl-long.730/) . As it is a dataset, it is essential to have some human evaluation to assess the quality.

B. 025 to 031 and 574 to 582, correlation scale ranges from -1 to 1, please check if the improvement can be quantified in absolute percentages

C. Please mention the languages in the abstract

D. I am slightly skeptic about the results produced by fine-tuning. If similar translation methods are used for training and evaluation, it is likely that the model is getting biased towards synthetic data. I wonder if this is causing artificial gain (See https://arxiv.org/pdf/2201.13405.pdf, https://aclanthology.org/2020.emnlp-main.618/)
A quick check on the DuRecDial2.0 dataset may be helpful
https://aclanthology.org/2021.emnlp-main.356/

---

> ### Author Rebuttal · Authors · 2023-08-27
>
> **RQ1**. “The proposed dataset is an aggregation of different dialogue datasets focussing on different properties of dialogue - yet the emphasis is more on “coherence” and excludes other properties of dialogue evaluation.”
>
> - In the literature, “coherence” at the dialogue level [1-3] and “relevance” at the response level [4-6] are the most evaluated dimensions. **The focus of our paper is to examine the evaluation capability of automatic metrics on multilingual dialogues rather than multi-dimensional dialogue evaluation [7-9]**. Hence, it is reasonable to select these two dimensions for analysis so that we can easily benchmark the performance of LLMs against previous state-of-the-art approaches.
>
> - Additionally, among the 18 English human-annotated dialogue datasets used in our paper, **coherence and relevance are the common dimensions shared by all the datasets**. On the other hand, collecting unified multi-dimensional annotations for all the 18 datasets and analyzing multi-dimensional evaluation capability by itself is a major problem, which we are also working on in progress. However, it is beyond the scope of the current paper.
>
> **RQ2**. “Further, the proposed dataset, despite being an evaluation benchmark for multilingual dialogue, uses automatic translation APIs and quality evaluation is done only on automatic metrics.”  and “It is disputed whether LLMs can be used for translation quality evaluation as zero-shot , further a lot of segment level evaluation metrics are not yet equipped to do quality evaluation (https://aclanthology.org/2023.acl-long.730/) . As it is a dataset, it is essential to have some human evaluation to assess the quality.”
>
> - First, we would like to emphasize that **we are working on multilingual automatic dialogue evaluation instead of machine translation**. MT is a tool for us to extend existing English-oriented dialogue data to other languages so that we can examine the multilingual dialogue evaluation capabilities of the BERT-based and LLM-based metrics.
>
> - Additionally, **we use a state-of-the-art commercial translation system** and the translation quality of our data is high as indicated by the different evaluation metrics. Also, **we use the most powerful LLM, GPT-4, for quality estimation. It has been shown to highly correlate with human evaluation on many NLP evaluation tasks [10-12]**.  Some researchers even use the output of GPT-4 to guide the development of NLG evaluation metrics [13].
>
> - **Besides GPT-4, we also use sacreBLEU, BERTScore, and BLEURT to assess the translation quality**. We compare the source sentence with the back-translated sentence using these metrics. As shown in Table 2 of the paper, all the metrics suggest that the translation quality of our dataset is good. **We perform additional quality estimation with Unbabel’s “wmt22-cometkiwi-da”**. Please find the results of all automatic metrics in the table below. We can see that **all the automatic metrics indicate high translation quality**, thereby suggesting that our benchmark is a valuable resource for both the dialogue and multilingual community.
>
>     | Metrics | EN-Zh | EN-ES | EN-DE | EN-FR | EN-JA | EN-KO | EN-HI | EN-AR | EN-RU |
>     |-----------|-----------|-----------|-----------|-----------|-----------|-----------|-----------|-----------|-----------|
>     | WMT22-cometkiwi-DA ↑ | 0.827 | 0.845 | 0.834 | 0.842 | 0.854  | 0.846 | 0.793 | 0.835 | 0.825 |
>     | GPT-4 ↑ | 4.58 | 4.64 | 4.63 | 4.59 | 4.33 | 4.26 | 4.40 | 4.39 | 4.40 |
>     | BLEU (with back-translation) ↑ | 37.85 | 56.58 | 50.30 | 52.10 | 42.70 | 40.23 | 49.33 | 48.54 | 49.13 |
>     | BLEURT (with back-translation) ↑ | 0.773 | 0.820 | 0.817 | 0.811 | 0.786 | 0.782 | 0.797 | 0.794 | 0.804 |
>     | BERTScore (with back-translation) ↑ | 0.691 | 0.767 | 0.754 | 0.748 | 0.711 | 0.707 | 0.726 | 0.737 | 0.741 |
>
> - **Additional human evaluation is performed**. Specifically, we collaborated with a service provider to have native speakers, proficient in English and their native language, evaluate the quality of translations from English to their mother tongue. Each reviewer assessed 350 translation pairs, rating them on a 1-5 scale, where 1 is poor and 5 is excellent. These evaluations covered nine language pairs, totaling 3150 instances. A quality check was done on a random subset to ensure reliability. For the English-to-Chinese translations, three experts evaluated them. The inter-annotator agreement was 0.578, indicating medium agreement among experts. **We can see that the human evaluation results below reinforce that the quality of our translated data is good**.
>
>     | Metrics | EN-Zh | EN-ES | EN-DE | EN-FR | EN-JA | EN-KO | EN-HI | EN-AR | EN-RU |
>     |-----------|-----------|-----------|-----------|-----------|-----------|-----------|-----------|-----------|-----------|
>     | Human ↑ | 4.75 | 4.68 | 4.59 | 4.53 | 4.39  | 4.32 | 4.17 | 4.41 | 4.71 |
>
> **RQ3**. “The human annotation collected for computing the correlation is not clear - without this information, it is difficult to judge the usability of the dataset.”  and “How are the human annotations for evaluation obtained? What are the scores from the metrics compared against to compute the correlation?”
>
> - We extend existing English dialogue evaluation benchmark datasets to other languages with machine translation. **The original human annotations provided in each dataset are reused for the same responses or dialogues in other languages**.
>
> - All the relevant works of the 18 existing English datasets, which contain the human-annotation information, are cited in our paper. In fact, **most English dialogue evaluation datasets used in our study are well-known and widely adopted for benchmarking automatic dialogue evaluation metrics in existing literature**. Hence, descriptions of the human annotation information of the 18 datasets are supplementary and are not the main text of our work. In this paper, we are trying to comprehensive analyze how state-of-the-art BERT-based and LLM-based metrics perform on multilingual automatic dialogue evaluation.
>
> - **For the reviewer’s reference, we provide the human annotations information for Persona-USR [8] and Topical-USR [8] as an example**. Persona-USR [8] and Topical-USR [8] contain 300 and 360 dialogue context-response pairs respectively. Each response is annotated by three dialogue experts along six quality dimensions: Understandable (0-1), Natural (1-3), Contextual Relevance (1-3), Interesting (1-3), Uses Knowledge (0-1), Overall Quality (1-5). Note that the numbers in the bracket are the Likert scales. The inter-annotator agreements for quality dimensions of both datasets are generally good (> 0.5). For instance, the correlation score of Persona-USR is computed between metric scores assigned to the 300 context-response pairs and the average “Contextual Relevance” human-annotated scores of those pairs.
>
> **RQ4**. “025 to 031 and 574 to 582, correlation scale ranges from -1 to 1, please check if the improvement can be quantified in absolute percentages”
>
> - **Yes, it is a common practice in dialogue evaluation literature to report absolute improvements in terms of correlation scores [4]**. Existing dialogue evaluation metrics generally exhibit a positive correlation with human evaluation. During the human annotation process, we often use higher likert scale to indicate better dialogue or response quality.
>
> **RQ5**. “I am slightly skeptic about the results produced by fine-tuning. If similar translation methods are used for training and evaluation, it is likely that the model is getting biased towards synthetic data. I wonder if this is causing artificial gain (See https://arxiv.org/pdf/2201.13405.pdf, https://aclanthology.org/2020.emnlp-main.618/) A quick check on the DuRecDial2.0 dataset may be helpful https://aclanthology.org/2021.emnlp-main.356/”
>
> - First, the finetuning process is to adapt the LLMs to better understand the dialogue context. If the reviewer carefully checks Table 4, 5 and section 6.2, you may find that **when finetuning on the English synthetic dialogue data only, Alpaca-7B and Phoenix-7B also see improvements in both English and other languages compared to zero-shot evaluation, especially under the turn-level evaluation setting. This suggests that the improvements are due to the capturing of high-level dialogue information rather than translation artifacts**.
>
> - Second, **we use different translation models (not similar translation methods) to translate the synthetic training data and the test data respectively**. The test data is translated with high-quality commercial models while the synthetic data is translated by the open-source NLLB-200-3.3B model. The translation quality of NLLB-200-3.3B is obviously worse than the commercial model. Yet, even with a worse translation, finetuning leads to substantial gains over zero-shot evaluation (Table 5), suggesting that **the finetuning is actually robust against translation errors**.
>
> - Lastly, in the supplementary (Table 14 and section E in the paper), we also show **the performance of finetuned LLMs on natural human-annotated Chinese dialogues (data originally in Chinese, not translated from English)**. We can observe that the finetuned LLMs also perform quite well. For example, **Alpaca-7B finetuned on the multilingual synthetic data performs much better than zeroshot evaluation with ChatGPT**.
>
> - With the above reasons, **it is unlikely that there are artificial gains due to biases in the translation process**.
>
> _References_:
>
> [1] Mesgar, Mohsen, Sebastian Bücker, and Iryna Gurevych. "Dialogue Coherence Assessment Without Explicit Dialogue Act Labels." In Proceedings of the 58th Annual Meeting of the Association for Computational Linguistics, pp. 1439-1450. 2020.
>
> [2] Zhang, Chen, Yiming Chen, Luis Fernando D’Haro, Yan Zhang, Thomas Friedrichs, Grandee Lee, and Haizhou Li. "DynaEval: Unifying Turn and Dialogue Level Evaluation." In Proceedings of the 59th Annual Meeting of the Association for Computational Linguistics and the 11th International Joint Conference on Natural Language Processing (Volume 1: Long Papers), pp. 5676-5689. 2021.
>
> [3] Ghazarian, Sarik, Nuan Wen, Aram Galstyan, and Nanyun Peng. "DEAM: Dialogue Coherence Evaluation using AMR-based Semantic Manipulations." In Proceedings of the 60th Annual Meeting of the Association for Computational Linguistics (Volume 1: Long Papers), pp. 771-785. 2022.
>
> [4] Yi-Ting Yeh, Maxine Eskenazi, and Shikib Mehri. 2021. A Comprehensive Assessment of Dialog Evaluation Metrics. In The First Workshop on Evaluations and Assessments of Neural Conversation Systems, pages 15–33, Online. Association for Computational Linguistics.
> [5] Sai, Ananya B., Akash Kumar Mohankumar, Siddhartha Arora, and Mitesh M. Khapra. "Improving dialog evaluation with a multi-reference adversarial dataset and large scale pretraining." Transactions of the Association for Computational Linguistics 8 (2020): 810-827.
>
> [6] Zhang, Chen, Luis Fernando D'Haro, Qiquan Zhang, Thomas Friedrichs, and Haizhou Li. "PoE: A Panel of Experts for Generalized Automatic Dialogue Assessment." IEEE/ACM Transactions on Audio, Speech, and Language Processing 31 (2023): 1234-1250.
>
> [7] Zhang, Chen, Luis Fernando D’Haro, Qiquan Zhang, Thomas Friedrichs, and Haizhou Li. "FineD-Eval: Fine-grained Automatic Dialogue-Level Evaluation." In Proceedings of the 2022 Conference on Empirical Methods in Natural Language Processing, pp. 3336-3355. 2022.
>
> [8] Mehri, Shikib, and Maxine Eskenazi. "USR: An Unsupervised and Reference Free Evaluation Metric for Dialog Generation." In Proceedings of the 58th Annual Meeting of the Association for Computational Linguistics, pp. 681-707. 2020
>
> [9] Mehri, Shikib, and Maxine Eskenazi. "Unsupervised Evaluation of Interactive Dialog with DialoGPT." In Proceedings of the 21th Annual Meeting of the Special Interest Group on Discourse and Dialogue, pp. 225-235. 2020.
>
> [10] Liu, Yang, Dan Iter, Yichong Xu, Shuohang Wang, Ruochen Xu, and Chenguang Zhu. "Gpteval: Nlg evaluation using gpt-4 with better human alignment." arXiv preprint arXiv:2303.16634 (2023).
>
> [11] Naismith, Ben, Phoebe Mulcaire, and Jill Burstein. "Automated evaluation of written discourse coherence using GPT-4." In Proceedings of the 18th Workshop on Innovative Use of NLP for Building Educational Applications (BEA 2023), pp. 394-403. 2023.
>
> [12] Rathje, Steve, Dan-Mircea Mirea, Ilia Sucholutsky, Raja Marjieh, Claire Robertson, and Jay J. Van Bavel. "GPT is an effective tool for multilingual psychological text analysis." (2023).
>
> [13] Xu, Wenda, Danqing Wang, Liangming Pan, Zhenqiao Song, Markus Freitag, William Yang Wang, and Lei Li. "Instructscore: Towards explainable text generation evaluation with automatic feedback." arXiv preprint arXiv:2305.14282 (2023).

---

### Official Review · Reviewer_sT3H · 2023-08-06

**Soundness:** 2

**Excitement:**

2: Mediocre: This paper makes marginal contributions (vs non-contemporaneous work), so I would rather not see it in the conference.

**Paper Topic And Main Contributions:**

This paper establishes a benchmark for evaluating multi-lingual, multi-turn dialogues, creating a dataset that covers various languages to address the current lack of evaluation benchmarks for language. The study also establishes strong self-supervised and multilingual baselines.

**Reasons To Accept:**

This paper is well organized and clearly written.

**Reasons To Reject:**

This paper presents a benchmark for evaluating multilingual dialogues and creates a multi-round dialogue dataset in up to ten different languages, providing an excellent tool for assessing the multilingual ability of large models. However, in my opinion, the practicality and effectiveness of such evaluations in the field of multilingualism are not as useful and effective as other evaluation tools that examine the overall capabilities of large models. Therefore, the actual relevance of evaluating the multilingual ability of large models needs to be carefully considered, especially given that the multilingual ability of large language models is generally not poor nowadays.

**Reproducibility:**

3: Could reproduce the results with some difficulty. The settings of parameters are underspecified or subjectively determined; the training/evaluation data are not widely available.

**Reviewer Confidence:**

3: Pretty sure, but there's a chance I missed something. Although I have a good feel for this area in general, I did not carefully check the paper's details, e.g., the math, experimental design, or novelty.

---

> ### Author Rebuttal · Authors · 2023-08-25
>
> **RQ1**. "The practicality and effectiveness of such evaluations in the field of multilingualism are not as useful and effective as other evaluation tools that examine the overall capabilities of large models."
>
> - Although there are other tools evaluating the overall capabilities of LLMs, **none of them specifically focus on dialogue, nor evaluate their capabilities for automatic assessments of dialogues**.
>
> - In the dialogue community, **automatic dialogue evaluation remains an important and open problem [1-5]**. Table 6 in the limitation section reveals that even GPT-4  (by far the most powerful LLM), doesn't perfectly align with human judgment on the FED-Dial benchmark (> 0.8), indicating a need for further exploration in LLMs' dialogue evaluation capabilities.
>
> - **Our paper complements existing LLM evaluation works and provides an important analysis on how existing LLMs perform automatic dialogue evaluation. We also examine their abilities to evaluate dialogues in the multilingual settings**. The evaluation of multilingual dialogues is also attracting significant interest. For example, the recent DSTC11 challenge (https://dstc11.dstc.community/) has one shared task dedicated to this topic. **The resources we introduced are important contributions to the NLG community**.
>
> **RQ2**. "The actual relevance of evaluating the multilingual ability of large models needs to be carefully considered, especially given that the multilingual ability of large language models is generally not poor nowadays"
>
> - We want to emphasize that **our works target automatic evaluation of multi-lingual dialogue, which is not just examining the multilingual ability of LLMs**. We also test their capabilities in understanding the dialogue contexts in a multilingual setting thereby providing accurate assessment.
>
> - Additionally, even though the multilingual ability of large language models is generally not poor, if the reviewer carefully checks existing LLMs, especially the open-source LLMs, **most of them are targeting popular languages such as English or Chinese**, however their performance in other languages is under-explored.
>
> - Lastly, current dialogue evaluation datasets are english-oriented, in this paper **we have also made an important effort to have a unified resource in 10 different languages**.
>
> _References_:
>
> [1] Yi-Ting Yeh, Maxine Eskenazi, and Shikib Mehri. 2021. A Comprehensive Assessment of Dialog Evaluation Metrics. In The First Workshop on Evaluations and Assessments of Neural Conversation Systems, pages 15–33, Online. Association for Computational Linguistics.
>
> [2] Huynh, Jessica, Cathy Jiao, Prakhar Gupta, Shikib Mehri, Payal Bajaj, Vishrav Chaudhary, and Maxine Eskenazi. "Understanding the Effectiveness of Very Large Language Models on Dialog Evaluation." arXiv preprint arXiv:2301.12004 (2023).
>
> [3] Gupta, Prakhar, Cathy Jiao, Yi-Ting Yeh, Shikib Mehri, Maxine Eskenazi, and Jeffrey P. Bigham. "InstructDial: improving zero and few-shot generalization in dialogue through instruction tuning." In Proceedings of the 2022 Conference on Empirical Methods in Natural Language Processing, pp. 505-525. 2022.
>
> [4] Eric Smith, Orion Hsu, Rebecca Qian, Stephen Roller, Y-Lan Boureau, and Jason Weston. 2022. Human Evaluation of Conversations is an Open Problem: comparing the sensitivity of various methods for evaluating dialogue agents. In Proceedings of the 4th Workshop on NLP for Conversational AI, pages 77–97, Dublin, Ireland. Association for Computational Linguistics.
>
> [5] Mehri, Shikib, Jinho Choi, Luis Fernando D'Haro, Jan Deriu, Maxine Eskenazi, Milica Gasic, Kallirroi Georgila et al. "Report from the nsf future directions workshop on automatic evaluation of dialog: Research directions and challenges." arXiv preprint arXiv:2203.10012 (2022).
>
> [6] Mehri, Shikib, and Maxine Eskenazi. "Unsupervised Evaluation of Interactive Dialog with DialoGPT." In 21th Annual Meeting of the Special Interest Group on Discourse and Dialogue, p. 225. 2020.

---

### Meta-Review · Area_Chair_2bYt · 2023-09-19

**Recommendation:** 3

**Metareview:**

This paper presents xDial-Eval, an evaluation benchmark for multilingual open-domain dialogue systems. In addition to its contribution as a new evaluation resource, the paper also shared insights into applying LLMs to automatic dialogue evaluation.

Two reviewers pointed out the need for human evaluation on the automatic machine translation quality during the construction of the multilingual dataset, to which the authors' rebuttal provided some further details on the conducted human evaluation results. However, given these new results were missing from the original paper submission, and somewhat in contradict with the description under section 3,
they are not considered as sufficient to override to the original review criticism.

One of the reviews was not taken into consideration for the AC recommendation due to the lack of details and no reviewer response during rebuttal period.

---

### Meta-Review · Senior_Area_Chairs · 2023-10-05

**Recommendation:** 3

**Metareview:**

meta review

---

### Decision · Program_Chairs · 2023-10-07

**Decision:**

Accept-Findings

**Comment:**

This paper presents xDial-Eval, an evaluation benchmark for multilingual open-domain dialogue systems. In addition to its contribution as a new evaluation resource, the paper also shared insights into applying LLMs to automatic dialogue evaluation.

Two reviewers pointed out the need for human evaluation on the automatic machine translation quality during the construction of the multilingual dataset, to which the authors' rebuttal provided some further details on the conducted human evaluation results. However, given these new results were missing from the original paper submission, and somewhat in contradict with the description under section 3,
they are not considered as sufficient to override to the original review criticism.

One of the reviews was not taken into consideration for the AC recommendation due to the lack of details and no reviewer response during rebuttal period.|meta review